# BIG5-CHAT: SHAPING LLM PERSONALITIES THROUGH TRAINING ON HUMAN-GROUNDED DATA

## ABSTRACT

In this work, we tackle the challenge of embedding realistic human personality traits into LLMs. Previous approaches have primarily focused on prompt-based methods that describe the behavior associated with the desired personality traits, suffering from realism and validity issues. To address these limitations, we introduce BIG5-CHAT, a large-scale dataset containing 100,000 dialogues designed to ground models in how humans *express* their personality in text. Leveraging this dataset, we explore Supervised Fine-Tuning and Direct Preference Optimization as training-based methods to align LLMs more naturally with human personality patterns. Our methods outperform prompting on personality assessments such as BFI and IPIP-NEO, with trait correlations more closely matching human data. Furthermore, our experiments reveal that models trained to exhibit higher conscientiousness, higher agreeableness, lower extraversion, and lower neuroticism display better performance on reasoning tasks, aligning with psychological findings on how these traits impact human cognitive performance. To our knowledge, this work is the first comprehensive study to demonstrate how training-based methods can shape LLM personalities through learning from real human behaviors.

## 1 INTRODUCTION

Realistically simulating human personality and its impact on text generation is a challenging yet crucial problem (Elster, 2015; Park et al., 2023; Serapio-García et al., 2023; Li et al., 2024; Frisch & Giulianelli, 2024). Embedding personality traits into LLMs can greatly enhance their authenticity across a wide range of applications, from conversational agents (Pradhan & Lazar, 2021) to educational tools (Kanero et al., 2022) and mental health platforms (Tudor Car et al., 2020; Ahmad et al., 2022). By creating more human-like interactions, LLMs can better simulate diverse personas and adapt more reliably to different contexts (Gao et al., 2024a).

However, existing methods primarily rely on prompting models with descriptions of behaviors associated with personality traits (e.g., "You are the life of the party"; Mao et al., 2023; Chen et al., 2024b; 2022; Tu et al., 2024). These behavior descriptions are often drawn from the same psychological questionnaires used to test their personality, raising evaluation validity concerns. More importantly, these behavioral descriptions are nonsensical for text-based LLMs (LLMs do not attend parties), failing to ground their personality in realistic patterns of how humans' personality is expressed in text (Vu et al., 2024). Additionally, the scarcity of large-scale, human-generated datasets annotated with personality traits has hindered the exploration of training-based approaches, limiting most prior research to prompting-based methods.

In this work, we address the challenge of inducing realistic human personality traits in LLMs by constructing a large-scale dialogue dataset, BIG5-CHAT, which is grounded in real human personality expressions in text. The overview of our work is illustrated in Figure 1. We choose the well-known Big Five personality traits framework to study this (McCrae & John, 1992; Pittenger, 1993), due to its reliability and validity as shown from psychological research. While previous datasets typically include only persona descriptions, our dataset bridges the gap between narrow-domain personality data and general-domain social interactions, ensuring both authenticity and scenario diversity. To achieve this, we combine two primary data sources — PsychGenerator (Vu et al., 2024), a collection of 850K Facebook posts annotated with Big Five trait scores, and SODA (Kim et al., 2022), a rich dataset of diverse social interactions — by utilizing product-of-experts text generation (DExperts;

Figure 1: Overview of the PSYCHSTEER method and evaluation. The expert generator was trained on the PsychGenerator dataset to induce Big Five personality traits (Vu et al., 2024) and integrated with the base model using the Dexperts framework alongside SODA's social scenarios (Liu et al., 2021; Kim et al., 2023a) to generate the BIG5-CHAT dataset. Various alignment methods were then evaluated for their effectiveness in inducing personality and their impact on reasoning benchmarks.

Liu et al., 2021). This combination enables us to capture the nuanced expression of personality traits across a wide range of dialogue scenarios.

Leveraging our BIG5-CHAT dataset, we empirically investigate how training-based methods grounded in real human data compare to traditional prompting techniques for inducing personality traits in LLMs, including instruction-based and demonstration-based prompting. Specifically, we explore Supervised Fine Tuning (SFT) and Direct Preference Optimization (DPO) (Rafailov et al., 2024) to align LLMs' personalities with Big Five traits. This comparison is crucial for understanding whether data-driven training methods can offer deeper, more reliable personality integration than the surface-level traits typically induced through prompting. Our results demonstrate that both SFT and DPO outperform prompting on two widely recognized Big Five personality tests: the BFI (John et al., 1999) and IPIP-NEO (Johnson, 2014).

In humans, personality traits often correlate with reasoning abilities (John et al., 1999; Soto et al., 2011), raising the question of how embedding personality traits in LLMs may influence their reasoning performance. To explore this, we evaluate our aligned models not only with traditional personality tests but also across five reasoning domains: social reasoning using SocialIQA (Sap et al., 2019), math reasoning using GSM8K (Cobbe et al., 2021) and MathQA (Amini et al., 2019), hallucination detection using TruthfulQA (Lin et al., 2021), commonsense reasoning using CommonsenseQA (Talmor et al., 2019) and PIQA (Bisk et al., 2020), and general reasoning using MMLU (Hendrycks et al., 2020) and GPQA (Rein et al., 2023). Our experiments show that models trained with higher levels of conscientiousness and agreeableness consistently outperform others in reasoning tasks. Conversely, models with lower levels of extraversion and neuroticism exhibit improved reasoning performance in general. These findings mirror patterns between Big Five traits and different reasoning abilities observed in psychological studies in humans (Ackerman & Heggestad, 1997; Schaie et al., 2004), further demonstrating how our personality induction method embeds deeper psycholinguistic traits into models.

This work makes the following contributions:

- We introduce the first large-scale dataset, BIG5-CHAT [1], containing 100,000 dialogues across a wide spectrum of personality expressions, addressing the limitations of existing methods that rely on simple prompting without grounding in real human personality expressions in text;
- We perform quantitative evaluations comparing SFT and DPO to prompting in terms of imbuing LLMs with personality, showing that both training-based methods induce more pronounced personality traits and more realistic intra-trait correlations;
- We conduct comprehensive empirical investigations into how personality traits affect performance in both social reasoning and general reasoning tasks, revealing that LLMs with distinct personality traits demonstrate varying strengths and weaknesses across domains.

---

[1] Our dataset and code are uploaded to the submission system, and will be open-sourced upon acceptance.

## 2 BACKGROUND

Drawing from psychological research, the Big Five personality traits framework (McCrae & John, 1992; Pittenger, 1993), comprising five key factors—*Openness*, *Conscientiousness*, *Extraversion*, *Agreeableness*, and *Neuroticism*—has emerged as a reliable model for capturing LLM-simulated personality behavior (Karra et al., 2022; Serapio-García et al., 2023; Li et al., 2022; Pan & Zeng, 2023). According to Yarkoni (2010), *openness* reflects curiosity and a willingness to explore new ideas, which is expressed through a distinctive language style that includes frequent use of articles, prepositions, and words related to intellectual or cultural topics such as "poet" and "universe"; *conscientiousness*, associated with discipline, organization, and reliability, is marked by achievement-oriented language, characterized by terms like "completed" and the avoidance of impulsive language, including swear words; *extraversion*, characterized by sociability, assertiveness, and high energy, is associated with social and positive emotion words like "friends" and "drinking," highlighting social engagement; *agreeableness*, embodying compassion and cooperativeness with a focus of harmony relationships, is demonstrated through communal and affectionate language, such as "family" and "love," while avoiding negative terms; and *neuroticism*, linked to emotional instability and anxiety, is expressed by a higher frequency of negative emotion words, including anxiety, sadness, and anger. Compared to other personality models like the Myers-Briggs Type Indicator (MBTI), the Big Five offers greater reliability, validity, and empirical support, making it the preferred choice for personality research (McCrae & John, 1992; Pittenger, 1993). The MBTI, by contrast, has been criticized for its lack of scientific rigor, poor test-retest reliability, and questionable validity (Pittenger, 1993; Furnham, 1996). The Big Five model has been extensively validated across diverse cultures and populations, demonstrating high levels of consistency over time and predicting a wide range of life outcomes, such as job performance and mental health (McCrae & Costa Jr, 1997; John et al., 2008; Barrick & Mount, 1991; Soldz & Vaillant, 1999).

Various prompting approaches have been developed to induce Big Five personality traits in LLMs. They often employ pre-defined scripts or questionnaires to nudge the model towards expressing Big Five personality traits during interactions (Mao et al., 2023; Chen et al., 2024b; 2022; Tu et al., 2024). However, several challenges can arise from using prompting as the personality alignment approach:

**Lack of psycholinguistic depth**   LLMs with personalities induced directly through prompting often mirror only surface-level traits, lacking the psycholinguistic richness necessary for simulating authentic human behavior (Dorner et al., 2023; Sá et al., 2024; Olea et al., 2024). This is unsurprising, as capturing human-like psycholinguistic properties involves understanding dynamic human states shaped by ongoing social and environmental interactions (Bandura et al., 1961; Baldwin, 1992). Unlike LLMs, which generate responses based on static training data, humans continuously adjust their behaviors and communication styles through lived experiences and social feedback. This limitation makes LLMs less reliable when tasked with simulating nuanced human behavior on downstream tasks (Soni et al., 2023), which can lead to cariacture (Cheng et al., 2023).

**Validity concerns in personality induction and evaluation**   The prompts used to induce LLM personalities are often adapted from psychometric questionnaires (Jiang et al., 2023; Tan et al., 2024), which could also be used later to assess the same personality traits. This dual use of questionnaires for both personality induction and evaluation raises concerns about validity (Lievens et al., 2007), and lead to biased assessments that do not accurately reflect generalization capabilities (Serapio-García et al., 2023; Xu et al., 2024). This issue becomes particularly problematic in downstream tasks, where the models designed this way are prone to overfitting to specific linguistic features rather than adapting robustly to diverse real-world contexts (Mizrahi et al., 2024). Thus, there is a need for more robust methods that can decouple the induction and evaluation processes.

**Unintended influence on reasoning patterns**   Role-based prompting may significantly influence LLM behavior and reasoning patterns, introducing the risk of altering the model's decision-making approach in unintended ways (Zheng et al., 2023). While this influence is not inherently negative, the responses of LLMs with personality prompting can be disproportionately shaped by the sparse, explicitly specified features of the prompt (Lu et al., 2021; Sclar et al., 2023). As a result, their behavior in reasoning tasks may be overly narrow, reflecting only the traits highlighted in the prompt rather than engaging a broader spectrum of cognitive strategies. This can lead to unexpected or

imbalanced responses, particularly in contexts where the model's reasoning should involve more comprehensive or nuanced thinking.

## 3 METHODOLOGY

The lack of large-scale datasets featuring personality-grounded dialogues poses a significant challenge. To address this challenge, we combine controllable text generation models with a domain-specific, personality-annotated dataset. Specifically, we utilize the DExperts framework (Liu et al., 2021) and the PsychGenerator dataset (Vu et al., 2024) to create BIG5-CHAT, a novel dataset that encapsulates diverse personality expressions within rich dialogue scenarios. The DExperts framework allows us to guide the language model's outputs toward specific personality traits during the generation process. Meanwhile, PsychGenerator provides a comprehensive collection of human-generated texts annotated with Big Five personality trait scores. By combining these technologies, we introduce PSYCHSTEER, an approach that effectively addresses the limitations of prior datasets by grounding personality traits in authentic human interactions.

### 3.1 DEXPERTS FRAMEWORK

DExperts allows us to control language model generation at decoding time by steering model outputs with expert generators. By integrating expert generators trained to exhibit different Big Five personality traits, we can induce personality within LLM outputs while maintaining dialogue quality. In the DExperts framework, let $M$ denote the pre-trained base language model, and $M^{\text{expert}}$ is the expert generator fine-tuned to generate text exhibiting the desired personality in our tasks. At each time step $t$, given the prompt and previous token sequence $x_{<t}$, the base model $M$ computes logits $z_t^{\text{base}} \in \mathbb{R}^{|V|}$, where $V$ is the vocabulary. The expert generator $M^{\text{expert}}$ computes logits $z_t^{\text{expert}}$ in the same manner. To integrate the influence of the expert generator, we adjust the base model's logits by incorporating the scaled difference between the expert generator model and base model logits:

$$z_t^{\text{combined}} = z_t^{\text{base}} + \gamma z_t^{\text{expert}}, \tag{1}$$

where $\gamma \in [0, +\infty)$ is a scaling factor controlling the degree of influence from the expert generator. This formulation effectively pulls the combined logits towards the expert generator logits, where $\gamma = 0$ results in using the base model's logits, and a larger $\gamma$ indicates a stronger influence of the expert generator's modification control. The combined logits $z_t^{\text{combined}}$ are transformed into a probability distribution, and the next token is sampled using the softmax function from this distribution.

### 3.2 EXPERT GENERATOR MODEL BASED ON SOCIAL MEDIA POSTS

To train expert generator models to exhibit certain personality traits, we perform SFT on the `LLaMA-3-8B-Instruct` model (Dubey et al., 2024) using the PsychGenerator dataset (Vu et al., 2024). This dataset comprises 846,304 Facebook posts, each paired with its author's Big Five personality trait scores. This dataset provides a robust foundation for training models to simulate nuanced human behaviors associated with different personality dimensions. We fine-tuned five expert generators, each representing and dedicated to generating text corresponding to one of the personality traits. For each personality trait, we converted the original floating-point trait labels into binary levels 'high'/'low' for each trait, allowing the distinct behaviors associated with the extreme ends of each trait to be more easily identified and analyzed.

We fine-tuned our expert generator models following the Alpaca format (Taori et al., 2023), which consists of three components: *instruction*, *input*, and *output*. In our training methodology:

- **Instruction**: We specify the name and level of a personality trait in the instruction. (e.g. *"Help me complete the sentence with certain Big Five Personality: Openness - high."*)
- **Input**: We provide the first five words of a post from the PsychGenerator dataset (e.g. *"who's got time to eat?"*). This serves as an initial context or prompt for the model.[2]

---

[2]We experimented with using only the first word as input. We empirically determined that using the first five words resulted in better generation quality.

- **Output**: The remainder of the post from the dataset (e.g. *"I'll just have a can of frosting."*), which typically embodies the specified personality trait.

When generating text completions with the PSYCHSTEER framework, the base model generates the first five words. This enables the expert generator model to influence the subsequent token generation by adjusting the logits to favor the desired personality trait while preserving coherence and fluency.

## 4 BIG5-CHAT DATASET

### 4.1 DATASET CONSTRUCTION

We introduce **BIG5-CHAT**, a large-scale dialogue responses dataset designed to capture Big Five personality traits within diverse social interactions. Our dataset construction leverages the SODA (Social DiAlogues) dataset (Kim et al., 2023a), which provides a diverse range of realistic social scenarios. SODA dialogues are generated by GPT-3.5 and enriched with social commonsense narratives, making it an ideal foundation for incorporating personality expressions due to its extensive coverage of social interactions. To induce personality traits into the dialogues, we employ the DExperts framework (Liu et al., 2021).

To build our dataset, we randomly sample 10,000 scenarios from SODA to provide diverse social contexts. In SODA, social interactions are modeled between two individuals referred to as Speaker X and Y, representing the participants in each dialogue. For each scenario, we generate a new utterance using our PSYCHSTEER framework to control for personality traits and get the dialogue responses between two participants. In the dialogues, one represents Speaker X (converted from the original SODA dialogue) and another represents Speaker Y with specific personality traits. For Speaker Y, based on the original responses from SODA, we generate new dialogue responses using the PSYCHSTEER framework. Examples of dialogues from our dataset are shown in Table 4. By conditioning on the preceding context (Speaker X's utterance), we use the base model $M$ guided by the expert generator $M^+$ specialized in the target personality trait to generate Speaker Y's responses. For each scenario, we generate pairwise dialogues by producing responses that reflect either high or low levels of the targeted personality trait. This approach results in pairs of dialogues that share the same context but differ in the expressed trait level. The process yields a total of 100,000 dialogues—20,000 for each trait, with an equal split between high and low trait levels.

### 4.2 DATASET STATISTICS

In this section, we examine the diversity and clarity of personality trait expressions within our BIG5-CHAT dataset. As illustrated in Table 4, we present examples where, for a single prompt from Speaker X, we have generated ten distinct responses from Speaker Y. These responses are conditioned on the high and low levels of each of the five Big Five personality traits. By varying only the level of a specific trait while keeping the prompt constant, we highlight how each personality trait distinctly influences conversational responses. Additionally, we analyze the token counts and other statistics of generated dialogue responses to ensure consistency across different personality trait levels in Table 5. The results indicate no significant differences in these statistics across different personality traits and levels, which suggests that the differences in statements are more related to content variations rather than spurious attributes such as context length. Further details about the dataset can be found in Appendix A.

Comparative analysis with existing personality datasets, as in Table 6, underscores several advantages of BIG5-CHAT. Unlike existing personality datasets such as Big5PersonalityEssays (Floroiu, 2024) and Machine-Mindset (Cui et al., 2023) which lack human-grounded data examples, our dataset consists of dialogues capturing dynamic and interactive conversational exchanges that are more representative of natural language use. Additionally, while previous work has focused solely on human-generated domain-specific data or synthetic machine-generated data, our approach combines both human dialogue and LLM to create realistic personality expressions. These findings are further validated through human evaluation, with more information available in Appendix C.1.

| Data Generation Method | Openness | Conscientiousness | Extraversion | Agreeableness | Neuroticism | Average |
|---|---|---|---|---|---|---|
| Test set | 93.7 | 94.2 | 93.4 | 93.4 | 94.3 | 93.8 |
| *Ours*: **Generator** | 82.5 | 80.0 | 80.0 | 81.0 | 78.5 | **80.4** |
| *Post-Completion*: `GPT-4o-mini` | 64.0 | 59.5 | 56.0 | 57.0 | 59.5 | 59.2 |
| *Topic-Post Generation*: `LLaMA-3-8B-Inst` | 66.0 | 73.0 | 81.0 | 88.5 | 83.0 | 78.2 |
| *Topic-Post Generation*: `GPT-4o-mini` | 65.0 | 78.0 | 80.0 | 85.5 | 84.0 | 78.5 |

Table 1: Accuracy (%) of the trained classifier in predicting each of the Big Five personality traits. The first row (Test set) shows the classifier's accuracy on the test split, demonstrating that the classifier is well-trained. The remaining rows display the performance of our generator model compared to two baselines, as assessed by the same classifier.

### 4.3 Evaluating Personality-Steering of the Data Generator

To help evaluate the quality of the generated dataset and its reflection of realistic personality traits, we trained a `RoBERTa-Large` (Liu et al., 2019) classifier with five regression heads using the MSE loss function. The model was trained on the PsychGenerator dataset, where the input consisted of text posts, and the output comprised the original trait labels, i.e., five floating-point values ranging from 0 to 1. The same train-validation-test split was applied here as with the expert generators. Training was conducted over five epochs with a learning rate of $1 \times 10^{-5}$. In Table 1, we observe that the classifier achieves an accuracy of 93.8% on the held-out test set, indicating that the PsychGenerator dataset contains distinct, learnable patterns that differentiate between high and low levels of personality traits.

Using the classifier as an evaluator, we demonstrate the high quality of the dataset generated by our expert generator, as shown at the bottom of Table 1, where it accurately reflects realistic personality traits. Specifically, we compare our dataset to two baselines for generating post datasets using LLMs: *Post-Completion* and *Topic-Post Generation*. *Post-Completion* replicates the expert generator's post generation strategy by prompting an LLM to complete a post given the first five words, the target personality traits, and the required post format for post-expression style guidance. *Topic-Post Generation*, on the other hand, is intentionally designed to be robust and prioritize performance over realism and controllability. It generates an entirely new post by first propmting an LLM to extract the main topic of a post from the PsychGenerator test set and then using one in-context post example to guide the LLMs in generating posts that match the desired personality traits, cover the extracted topic, and follow similar post-expression styles. We evaluated *Topic-Post Generation* using `GPT-4o-mini` (OpenAI, 2024) and *Post-Completion* using both `LLaMA-3-8B-Instruct` (Dubey et al., 2024) and `GPT-4o-mini` (OpenAI, 2024). For consistency, all experiments are based on the same set of 1,000 examples randomly chosen from the PsychGenerator test set. The classifier was used to evaluate the generated data by predicting the levels of each trait, and the quality was measured by whether the predictions matched the desired personality traits. Our results in Table 1 show that our expert generator outperforms both baselines, achieving higher average accuracy scores for every personality trait dimension compared to the *Post-Completion* baseline. Furthermore, it surpasses *Topic-Post Generation* when results are averaged across all traits. Additional details about the two baseline methods can be found in Appendix B.1. These findings are further validated through human evaluation, with more information available in Appendix C.2.

## 5 Experiments

In this section, we first outline the experimental setup in Section 5.1, detailing the training procedures for the expert generators and the evaluation of various alignment strategies used to induce personality traits in LLMs. Next, we present the results of the personality tests in Section 5.2, followed by an analysis of the models' reasoning performance in Section 5.3.

### 5.1 Experiment Setup

**Expert generator training**  We trained five expert generators, each dedicated to generating text corresponding to one of the Big Five personality traits. During training, we provided the instruction specifying the target binary level for the trait, enabling the generator to learn patterns characteristic of individuals with either high or low levels of the respective trait, as illustrated in Section 3.2.

| Method | Openness High ↑ | Low ↓ | Conscientiousness High ↑ | Low ↓ | Extraversion High ↑ | Low ↓ | Agreeableness High ↑ | Low ↓ | Neuroticism High ↑ | Low ↓ | Average High ↑ | Low ↓ |
|---|---|---|---|---|---|---|---|---|---|---|---|---|
| *BFI LLaMA-3-8B-Instruct* | | | | | | | | | | | | |
| Direct | 3.1 ± 0.1 | | 3.0 ± 0.0 | | 3.0 ± 0.0 | | 3.0 ± 0.0 | | 3.0 ± 0.0 | | 3.0 ± 0.0 | |
| Prompt-Inst | 5.0 ± 0.0 | 2.0 ± 0.3 | 4.9 ± 0.1 | 1.9 ± 0.1 | 4.8 ± 0.3 | 1.9 ± 0.1 | 4.9 ± 0.1 | 2.4 ± 0.4 | 4.1 ± 0.2 | 1.6 ± 0.0 | 4.7 ± 0.1 | 2.0 ± 0.2 |
| SFT | 5.0 ± 0.0 | 2.0 ± 0.2 | 5.0 ± 0.0 | 1.6 ± 0.1 | 4.7 ± 0.4 | 2.7 ± 0.5 | 5.0 ± 0.0 | 1.2 ± 0.1 | 4.1 ± 0.2 | 2.5 ± 0.0 | **4.8 ± 0.1** | 2.0 ± 0.2 |
| DPO | 5.0 ± 0.0 | 1.6 ± 0.2 | 5.0 ± 0.0 | 1.6 ± 0.1 | 4.8 ± 0.3 | 2.5 ± 0.0 | 4.8 ± 0.2 | 1.0 ± 0.0 | 3.5 ± 0.0 | 1.1 ± 0.1 | 4.6 ± 0.1 | **1.6 ± 0.1** |
| *BFI LLaMA-3-70B-Instruct* | | | | | | | | | | | | |
| Direct | 4.4 ± 0.1 | | 4.4 ± 0.1 | | 3.3 ± 0.1 | | 4.6 ± 0.1 | | 2.1 ± 0.2 | | 3.8 ± 0.1 | |
| Prompt-Demo | 4.0 ± 0.1 | 2.5 ± 0.1 | 4.0 ± 0.1 | 2.0 ± 0.1 | 4.5 ± 0.1 | 2.3 ± 0.1 | 4.4 ± 0.1 | 2.0 ± 0.0 | 3.6 ± 0.0 | 2.1 ± 0.1 | 4.1 ± 0.1 | 2.2 ± 0.1 |
| Prompt-Inst | 5.0 ± 0.1 | 1.8 ± 0.1 | 5.0 ± 0.1 | 1.6 ± 0.0 | 5.0 ± 0.0 | 1.4 ± 0.1 | 4.9 ± 0.0 | 1.5 ± 0.1 | 5.0 ± 0.1 | 1.6 ± 0.0 | **5.0 ± 0.0** | 1.6 ± 0.0 |
| SFT | 5.0 ± 0.0 | 1.2 ± 0.1 | 5.0 ± 0.1 | 1.4 ± 0.1 | 5.0 ± 0.0 | 1.2 ± 0.1 | 5.0 ± 0.1 | 1.6 ± 0.2 | 5.0 ± 0.0 | 1.1 ± 0.2 | **5.0 ± 0.0** | **1.3 ± 0.1** |
| DPO | 5.0 ± 0.0 | 1.5 ± 0.1 | 5.0 ± 0.0 | 1.5 ± 0.1 | 5.0 ± 0.0 | 1.0 ± 0.1 | 5.0 ± 0.0 | 1.8 ± 0.2 | 5.0 ± 0.0 | 1.1 ± 0.1 | **5.0 ± 0.0** | 1.4 ± 0.1 |
| *IPIP-NEO LLaMA-3-8B-Instruct* | | | | | | | | | | | | |
| Direct | 3.0 ± 0.1 | | 3.3 ± 0.0 | | 3.4 ± 0.1 | | 3.2 ± 0.0 | | 3.0 ± 0.1 | | 3.2 ± 0.1 | |
| Prompt-Inst | 4.4 ± 0.1 | 1.5 ± 0.1 | 4.5 ± 0.1 | 2.3 ± 0.1 | 5.0 ± 0.0 | 1.9 ± 0.0 | 4.6 ± 0.0 | 2.3 ± 0.1 | 4.2 ± 0.1 | 2.6 ± 0.1 | 4.5 ± 0.1 | 2.1 ± 0.1 |
| SFT | 4.3 ± 0.1 | 1.5 ± 0.1 | 4.5 ± 0.2 | 2.7 ± 0.1 | 5.0 ± 0.0 | 2.2 ± 0.1 | 4.0 ± 0.2 | 1.8 ± 0.2 | 4.3 ± 0.1 | 2.0 ± 0.1 | 4.4 ± 0.1 | **2.0 ± 0.1** |
| DPO | 5.0 ± 0.0 | 1.9 ± 0.1 | 5.0 ± 0.0 | 2.9 ± 0.1 | 5.0 ± 0.0 | 1.6 ± 0.1 | 4.5 ± 0.1 | 1.2 ± 0.0 | 3.8 ± 0.1 | 3.7 ± 0.1 | **4.7 ± 0.0** | 2.3 ± 0.1 |
| *IPIP-NEO LLaMA-3-70B-Instruct* | | | | | | | | | | | | |
| Direct | 3.6 ± 0.1 | | 4.0 ± 0.1 | | 3.5 ± 0.1 | | 4.0 ± 0.0 | | 2.3 ± 0.1 | | 3.5 ± 0.1 | |
| Prompt-Demo | 3.5 ± 0.0 | 2.5 ± 0.1 | 3.8 ± 0.0 | 2.2 ± 0.1 | 4.0 ± 0.1 | 2.5 ± 0.0 | 4.3 ± 0.0 | 2.1 ± 0.1 | 3.0 ± 0.1 | 2.2 ± 0.1 | 3.7 ± 0.0 | 2.3 ± 0.1 |
| Prompt-Inst | 4.6 ± 0.0 | 1.3 ± 0.0 | 5.0 ± 0.0 | 1.4 ± 0.0 | 5.0 ± 0.0 | 1.6 ± 0.0 | 4.8 ± 0.0 | 1.1 ± 0.1 | 4.9 ± 0.0 | 1.7 ± 0.1 | **4.9 ± 0.0** | 1.4 ± 0.0 |
| SFT | 4.9 ± 0.1 | 1.1 ± 0.0 | 5.0 ± 0.0 | 1.3 ± 0.1 | 5.0 ± 0.0 | 1.3 ± 0.0 | 4.9 ± 0.0 | 1.0 ± 0.0 | 4.9 ± 0.0 | 1.2 ± 0.1 | **4.9 ± 0.0** | **1.2 ± 0.0** |
| DPO | 4.8 ± 0.0 | 1.4 ± 0.1 | 5.0 ± 0.0 | 1.6 ± 0.1 | 5.0 ± 0.0 | 1.1 ± 0.1 | 4.9 ± 0.0 | 1.0 ± 0.0 | 5.0 ± 0.0 | 1.1 ± 0.0 | **4.9 ± 0.0** | 1.2 ± 0.1 |

Table 2: Personality test results for different alignment methods, demonstrating the greater effectiveness of training-based approaches in inducing Big Five personality traits. **Direct** refers to directly providing the test questions to the model without including personality-related prompts. **Prompt-Inst** refers to instruction-based prompting, and **Prompt-Demo** refers to demonstration-based prompting. Scores range from 1 to 5, where a score closer to 5 indicates stronger agreement with the trait, while a score closer to 1 reflects weaker or opposing agreement. The results for the other baselines are presented in Table 10.

For each trait, we fine-tuned a `LLaMA-3-8B-Instruct` model over one epoch using a learning rate of $1 \times 10^{-6}$. These fine-tuned models were subsequently used to produce expert-generated logits: $z_t^{\text{expert}}$. To create the BIG5-CHAT dataset, we used these expert generators in conjunction with a `LLaMA-3-70B-Instruct` model to generate $z_t^{\text{combined}}$ (Dubey et al., 2024), as described in Eq. (1). The scaling factor $\gamma$ was set to 0.5, and the dialogue data was generated using greedy decoding. More training details about the expert generator are explained in Appendix B.2.

**Prompting and training strategies** We implemented two baseline prompting strategies to induce personality traits in LLMs. The first strategy, *instruction-based prompting*, directly instructs the model to exhibit specific Big Five traits. The second strategy, *demonstration-based prompting*, involves providing the model with 10 in-context examples randomly selected from our BIG5-CHAT dataset to demonstrate the behaviors corresponding to the desired traits. The instruction-based approach relies on explicit descriptions (e.g., "what people typically do"), while the demonstration-based approach draws from behaviorally-driven examples (e.g., "what people typically say"). These baselines were compared to trained models using SFT and DPO, implemented via LoRA (Hu et al., 2022). These trained models were later prompted in a manner consistent with their training data format, where personality trait names and levels were explicitly specified in the instructions. The experiments were conducted using two versions of the LLaMA model: `LLaMA-3-8B-Instruct` and `LLaMA-3-70B-Instruct`. More prompting and training details are explained in Appendix B.3 and Appendix B.4.

**Evaluation procedure** For personality trait evaluation, we adopted the methodology from Huang et al. (2024) for the BFI test, which consists of 44 questions, each rated on a scale from 1 (strongly disagree) to 5 (strongly agree). For the IPIP-NEO test, we utilized the 120-question set from Jiang et al. (2024a), which also employed a 1 to 5 rating scale. We measured the standard deviation by repeating each experiment five times, using a temperature setting of 0.6. To assess reasoning capabilities, we evaluated the models across five domains: (1) social reasoning on SocialIQA (Sap et al., 2019), (2) math reasoning on GSM8K (Cobbe et al., 2021) and MathQA (Amini et al., 2019), (3) hallucination detection on TruthfulQA (Lin et al., 2021), (4) commonsense reasoning on CommonsenseQA (Talmor et al., 2019) and PIQA (Bisk et al., 2020), and (5) general reasoning on MMLU (Hendrycks et al., 2020) and GPQA (Rein et al., 2023). Further evaluation setup details are explained in Appendix B.5.

## 5.2 PERSONALITY TRAIT ASSESSMENT RESULTS

Table 2 presents the BFI and IPIP-NEO personality assessment results across direct inference and various alignment baselines and methods, including instruction-based prompting, demonstration-based prompting, SFT, and DPO. The performance trends are consistent across both personality tests. Compared to direct inference, which lacks any personality trait descriptions, both prompting and training methods successfully reflect the induced traits in their responses to the personality questionnaires. Specifically, these methods produce higher scores for high trait levels and lower scores for low trait levels, indicating that the traits are effectively embedded.

However, training-based methods, SFT and DPO, induce more pronounced personality traits than the two prompting-based approaches. Yet, we find no substantial difference between SFT and DPO. The training-based methods notably excel in producing lower scores for low levels of personality traits when compared to prompting-based methods. This highlights the efficacy of training on the BIG5-CHAT dataset to induce personality traits. In contrast, while demonstration-based prompting uses examples from the same dataset in context, it does not achieve similar results, likely due to the lack of explicit training. It is important to note that we excluded results for demonstration-based prompting on `LLaMA-3-8B-Instruct`, as the model exhibited a significant decline in instruction-following performance, making it difficult to extract meaningful answers. Overall, the `LLaMA-3-8B-Instruct` model underperforms compared to `LLaMA-3-70B-Instruct`, which is expected given the difference in parameter size and instruction-following capabilities. Further details regarding the assessment of personality traits can be found in Appendix C.3.

In addition, to evaluate how effectively the prompting and training methods replicate the intra-trait correlations observed in human data, we calculated these correlations using real human distributions derived from the IPIP-NEO questionnaire. Our results indicate that the training models, particularly those using SFT, more accurately capture the trait correlations found in natural human data compared to prompting-based methods. Further details on the intra-trait correlations can be found in Appendix C.4.

## 5.3 REASONING EVALUATION RESULTS

The reasoning evaluation results for our training methods and baselines are shown in Table 3 for `LLaMA-3-70B-Instruct` and in Table 12 for `LLaMA-3-8B-Instruct`, covering five reasoning domains. Overall, SFT consistently outperformed or matched DPO for the 70B model. This indicates that training on BIG5-CHAT does not impair question-answering abilities; in fact, training, especially with SFT, enhances social, mathematical, and commonsense reasoning for specific personality traits compared to direct inference. When comparing trait levels, models with higher conscientiousness and agreeableness generally outperformed those with lower levels. Openness showed no clear performance difference between levels, while models simulating lower levels of extraversion and neuroticism performed better. These trends were consistent across the majority of the benchmarks, indicating that certain personality trait levels can improve performance in reasoning tasks. Additional results and analyses for both models are provided in Appendix C.5 and Appendix C.6.

Furthermore, existing psychological research on the Big Five personality traits shows that openness, conscientiousness, and agreeableness enhance reasoning abilities for humans, while neuroticism and extraversion tends to impair cognition (John et al., 1999; Soto et al., 2011; Ackerman & Heggestad, 1997; Schaie et al., 2004; Chamorro-Premuzic et al., 2006). The differences in performance across traits on reasoning benchmarks in our study somewhat align with these findings, as summarized in Table 13, and reflect patterns observed in human problem-solving and reasoning tasks (Ackerman & Heggestad, 1997; Schaie et al., 2004). Specifically, both the performance of `LLaMA-3-70B-Instruct` and evidence from psychological studies suggest that higher levels of conscientiousness and agreeableness, and lower levels of extraversion and neuroticism, are associated with improved reasoning outcomes. However, while high openness is beneficial for human cognition, the model does not exhibit significant gains in reasoning tasks beyond math. This divergence between human and model performance suggests that the influence of openness on reasoning in large language models might be domain-specific or limited in scope. A more detailed discussion on the correlation between personality traits and reasoning behaviors can be found in Appendix D.1 for the 70B model, and in Appendix D.2 for the 8B model.

| Benchmark | Direct | Method | Openness | | Conscientiousness | | Extraversion | | Agreeableness | | Neuroticism | | Average | |
|---|---|---|---|---|---|---|---|---|---|---|---|---|---|---|
| | | | High↑ | Low↑ | High↑ | Low↑ | High↑ | Low↑ | High↑ | Low↑ | High↑ | Low↑ | High↑ | Low↑ |
| *Social Reasoning* | | | | | | | | | | | | | | |
| SocialIQA | 46.6 | Prompt | 40.8 | 43.9 | 42.9 | 39.9 | 43.3 | 42.0 | 42.4 | 40.8 | 39.1 | 44.1 | 41.7 | 42.1 |
| | | SFT | 50.3 | 50.4 | 50.9 | 46.8 | 50.0 | 50.3 | 50.5 | 46.6 | 48.2 | 50.6 | 50.0 | 48.9 |
| | | DPO | 41.5 | 44.5 | 44.7 | 37.6 | 43.0 | 43.6 | 44.8 | 39.0 | 40.0 | 45.3 | 42.8 | 42.0 |
| *Math Reasoning* | | | | | | | | | | | | | | |
| GSM8K | 80.6 | Prompt | 75.7 | 70.1 | 73.5 | 32.6 | 80.8 | 33.5 | 87.2 | 77.8 | 26.0 | 89.4 | 68.6 | 60.7 |
| | | SFT | 85.8 | 76.2 | 86.4 | 81.7 | 85.1 | 86.7 | 87.0 | 74.5 | 76.0 | 87.3 | 84.1 | 81.3 |
| | | DPO | 87.9 | 88.5 | 90.2 | 80.6 | 88.9 | 90.4 | 87.3 | 90.0 | 15.2 | 91.0 | 73.9 | 88.1 |
| MathQA | 39.0 | Prompt | 33.5 | 33.5 | 32.8 | 31.5 | 32.3 | 33.3 | 33.6 | 32.4 | 32.1 | 34.1 | 32.9 | 33.0 |
| | | SFT | 43.3 | 42.6 | 43.0 | 43.3 | 43.2 | 42.7 | 42.9 | 42.9 | 42.8 | 43.3 | 43.0 | 43.0 |
| | | DPO | 33.9 | 34.7 | 32.9 | 28.1 | 30.5 | 35.0 | 31.3 | 32.8 | 28.9 | 34.0 | 31.5 | 32.9 |
| *Hallucination Detection* | | | | | | | | | | | | | | |
| TruthfulQA | 58.6 | Prompt | 54.1 | 51.1 | 55.9 | 45.2 | 52.0 | 55.7 | 52.3 | 49.1 | 48.9 | 58.6 | 52.6 | 51.9 |
| | | SFT | 55.2 | 52.8 | 55.6 | 50.8 | 54.5 | 56.7 | 54.4 | 51.6 | 52.4 | 56.7 | 54.4 | 53.7 |
| | | DPO | 54.6 | 54.2 | 64.6 | 38.5 | 46.0 | 65.3 | 59.6 | 50.6 | 43.0 | 65.8 | 53.6 | 54.9 |
| *Commonsense Reasoning* | | | | | | | | | | | | | | |
| CommonsenseQA | 27.0 | Prompt | 60.0 | 59.9 | 22.5 | 22.3 | 35.5 | 50.0 | 45.0 | 34.9 | 20.2 | 36.8 | 36.6 | 40.8 |
| | | SFT | 77.7 | 78.8 | 77.6 | 66.0 | 75.7 | 78.9 | 77.0 | 73.8 | 79.1 | 78.5 | 77.4 | 75.2 |
| | | DPO | 57.7 | 65.9 | 23.8 | 25.8 | 23.2 | 70.8 | 21.3 | 39.2 | 20.1 | 44.6 | 29.2 | 49.3 |
| PIQA | 80.4 | Prompt | 79.6 | 79.8 | 80.5 | 77.3 | 78.0 | 80.0 | 79.8 | 78.4 | 78.8 | 80.7 | 79.3 | 79.2 |
| | | SFT | 81.2 | 81.0 | 81.2 | 80.4 | 81.8 | 81.3 | 81.2 | 80.0 | 81.0 | 81.2 | 81.3 | 80.8 |
| | | DPO | 76.4 | 76.8 | 79.4 | 70.9 | 76.4 | 79.8 | 78.5 | 74.0 | 72.9 | 79.5 | 76.7 | 76.2 |
| *General Reasoning* | | | | | | | | | | | | | | |
| MMLU | 74.5 | Prompt | 70.3 | 69.6 | 40.6 | 52.8 | 56.9 | 72.8 | 69.0 | 69.2 | 55.3 | 67.9 | 58.4 | 66.5 |
| | | SFT | 72.5 | 72.0 | 73.1 | 68.6 | 72.1 | 73.5 | 72.8 | 70.7 | 72.5 | 73.8 | 72.6 | 71.7 |
| | | DPO | 57.9 | 64.4 | 50.3 | 33.8 | 42.3 | 72.3 | 34.3 | 62.5 | 33.2 | 69.1 | 43.6 | 60.4 |
| GPQA | 33.5 | Prompt | 31.5 | 34.2 | 31.7 | 32.4 | 34.6 | 32.8 | 32.4 | 32.5 | 31.9 | 32.1 | 32.4 | 32.7 |
| | | SFT | 33.5 | 32.4 | 34.2 | 34.2 | 33.3 | 34.4 | 33.3 | 33.3 | 34.4 | 33.5 | 33.7 | 33.6 |
| | | DPO | 36.8 | 31.9 | 35.7 | 30.6 | 35.9 | 35.9 | 35.5 | 35.7 | 32.6 | 34.6 | 35.3 | 33.7 |
| **Average** | 55.0 | Prompt | 55.7 | 55.3 | 47.6 | 41.8 | 51.7 | 49.9 | 55.2 | 51.9 | 41.5 | 55.5 | 50.3 | 50.9 |
| | | SFT | 62.4 | 60.8 | 62.7 | 59.0 | 62.0 | 63.1 | 62.4 | 59.2 | 60.8 | 63.1 | 62.1 | 61.0 |
| | | DPO | 55.8 | 57.6 | 52.7 | 43.2 | 48.3 | 61.6 | 49.1 | 53.0 | 35.7 | 58.0 | 48.3 | 54.7 |

Table 3: Benchmark results for different personality traits on `LLaMA-3-70B-Instruct`. The evaluation metrics and full experiment results including standard deviations are detailed in Appendix C.5. **Direct** refers to direct inference without including personality-related prompts. **Prompt** refers to instruction-based prompting. On average, SFT achieves the best performance. Higher levels of conscientiousness and agreeableness, along with lower levels of extraversion and neuroticism, generally enhance reasoning capabilities.

# 6 RELATED WORKS

## 6.1 INDUCING PERSONALITY TRAITS IN LLMS

The personality traits of LLMs greatly influence their responses to human prompts, making personality alignment a key research area(Chen et al., 2024b; Jiang et al., 2024b; Kovačević et al., 2024; Lee et al., 2024; Zhu et al., 2024; Anthropic, 2024). Approaches include parameter-frozen methods, like in-context learning and retrieval-augmented generation, which configure personality profiles within the context of interactions without altering model parameters (Chen et al., 2022; Jiang et al., 2024a; Tu et al., 2024), and parameter-tuning methods, such as supervised fine-tuning, RLHF, and DPO, which adjust model parameters to internalize personality traits (Petrov et al., 2024; Vu et al., 2024; Stiennon et al., 2020; Ouyang et al., 2022; Zhang et al., 2024; Zeng et al., 2024b;a). While many studies use LLM-generated data to induce personality traits, these texts often lack human-like psycholinguistic properties (Cui et al., 2023; Chen et al., 2024a; Muñoz-Ortiz et al., 2023; Seals & Shalin, 2023). In contrast, our work utilizes an expert generator model trained on real human data with specific Big Five traits to guide alignment data generation, offering a more human-like approach to inducing personality traits in LLMs.

## 6.2 ASSESSING PERSONALITY TRAITS IN LLMS

Various psychological theories, particularly the Big Five model, have played a key role in understanding human personality traits, examining dimensions such as openness, conscientiousness, extraversion, agreeableness, and neuroticism (Cattell, 1957; Myers et al., 1962; John et al., 1999; Paulhus & Williams, 2002; Sato, 2005). These traits are often measured using psychometric tests like the Big Five Inventory (BFI) (John et al., 1999) and the NEO-PI-R (Costa & McCrae, 2008). In

recent studies, similar assessments have been adapted to LLMs using prompting techniques (Huang et al., 2024; Karra et al., 2022; Petrov et al., 2024). However, the validity and reliability of these methods remain contested (Shu et al., 2024; Huang et al., 2023; Serapio-García et al., 2023). Our approach builds on this work by evaluating the personalities of LLMs post-alignment using a zero-shot classifier and testing their capabilities on social and general reasoning benchmarks, demonstrating the effectiveness of our alignment method (Tan et al., 2024; Kim et al., 2023b; Zhu et al., 2024).

## 7 CONCLUSION

In this work, we addressed the challenge of embedding realistic human personality traits into LLMs by introducing BIG5-CHAT, a large-scale dataset of 100,000 dialogues capturing realistic Big Five personality expressions. Previous prompting-based approaches often exaggerated traits and raised validity concerns, so we used SFT and DPO on BIG5-CHAT to induce personality traits more naturally. Our results show that these training-based methods outperform prompting on personality assessments such as BFI and IPIP-NEO, with more expressive and pronounced traits and intra-trait correlations that align with human data. Furthermore, we observed that LLMs trained with higher levels of conscientiousness and agreeableness excel in various reasoning tasks, including social, mathematical, commonsense, general reasoning, and hallucination detection, while models with lower extraversion and neuroticism performed better at all reasoning tasks. These findings align with psychological studies on personality's impact on human cognition. Our work demonstrates that training-based approaches grounded in real human data can more effectively shape LLM personalities and improve reasoning performance, offering a novel pathway for developing adaptive, human-like AI systems.

## 8 LIMITATIONS & FUTURE WORK

While our study aims to embed realistic human personality traits into LLMs, there are several limitations that can be addressed in future work. First, our focus on the Big Five personality traits, while well-established, may not capture the full spectrum of human personality. Other frameworks, such as Dark Triad Dirty Dozen (Jonason & Webster, 2010) and EPQ-R (Eysenck, 1997), could provide additional insights into the generalizability of personality induction in LLMs. Second, there is a risk of inadvertently reinforcing societal biases, as LLMs trained on human-generated data may inherit harmful stereotypes or undesirable behaviors (Kotek et al., 2023; Liao & Wortman Vaughan, 2024). Although our induced personalities are intended to be neutral, further research is needed to ensure LLMs do not replicate or amplify biases or abnormal mental behaviors, which could negatively impact their usage. Third, while our study investigates the correlation between personality traits and reasoning capabilities, this analysis is limited to specific tasks and contexts. Expanding this research to include a broader range of reasoning tasks and scenarios would provide a deeper understanding of how different traits influence cognitive abilities in LLMs. Finally, our current approach isolates individual traits for steering, but personality traits are rarely exhibited in isolation. Although our method is naturally extensible to multi-trait steering by combining logits from multiple expert models during decoding, we deliberately focus on single traits in this study to enhance clarity, interpretability, and replicability, consistent with established practices in personality modeling research (Jiang et al., 2023). Nevertheless, multi-trait interactions are an important area for future exploration. Extending our approach to steer multiple traits simultaneously could enable the generation of more complex, blended personality profiles and provide deeper insights into the interconnectedness of traits. These limitations highlight important areas for future exploration in creating more nuanced, ethical, and effective personality-imbued LLMs.

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

## A    ADDITIONAL BIG5-CHAT DATASET STATISTICS

The SODA dataset spans a wide range of topics commonly encountered in social interactions (Kim et al., 2023a). It captures diverse emotional nuances such as curiosity and disappointment, alongside thematic elements related to attributes, effects, intentions, needs, reactions, and wants. This extensive variety makes the BIG5-CHAT dataset a valuable resource for analyzing complex conversational contexts and emotional dynamics. Its broad coverage enhances the generalizability of models trained on this data, enabling them to handle diverse social scenarios effectively.

Table 4 presents example conversations from the BIG5-CHAT dataset, illustrating how Speaker Y's responses vary according to different levels of the Big Five personality traits. Each section showcases the influence of high and low levels of Openness, Conscientiousness, Extraversion, Agreeableness, and Neuroticism on conversational style. These examples highlight the nuanced ways in which personality dimensions shape conversational dynamics and response patterns, even within identical situational contexts.

A statistical analysis of the dataset is provided in Table 5, detailing metrics such as token count, sentence count, vocabulary size, sentence length, and total vocabulary diversity. These statistics reveal linguistic patterns associated with varying levels of personality traits. For instance, conversations with higher levels of Openness and Extraversion tend to feature longer sentences and larger vocabularies, reflecting a richer and more elaborate expression style. In contrast, conversations tied to lower levels of these traits exhibit shorter, more concise sentence structures and less vocabulary diversity, indicating a simpler and more focused communication style.

Table 6 provides a comparative analysis of the BIG5-CHAT dataset against other prominent personality datasets. The comparison highlights key aspects such as the personality framework employed, the realism of personalities (i.e., whether generated by humans or LLMs), dataset size, interaction types, and the alignment methods used. This overview emphasizes the distinctive features and strengths of the BIG5-CHAT dataset, underscoring its unique contributions to personality-related research compared to existing resources.

## B    ADDITIONAL IMPLEMENTATION DETAILS

### B.1    DETAILS OF BASELINES FOR EVALUATING THE EXPERT GENERATOR

**Baseline 1: Post-Completion**    The following prompt was used for the *Post-Completion* baseline of `GPT-4o-mini`, as referenced in Table 1:

```
Here is an example of Facebook posts: {an_example_post}
Help me complete the sentence with certain Big Five Personality
following the Facebook post format: {trait} - {level}
Sentence: {first_five_words}
Directly provide the completed Facebook post according to the
requirements without any explanations.
```

The example post was randomly selected from the PsychGenerator test set but is not part of the 1,000 examples. We used greedy decoding to prompt the LLMs. This baseline is designed to closely replicate the generation process of the expert generator. Specifically, it generates a completion for a post using only the first five words of the input and the embedded personality traits.

**Baseline 2: Topic-Post Generation**    The following prompt was used as the baseline for *Topic-Post Generation* with `GPT-4o-mini` and `LLaMA-3-8B-Instruct`, as referenced in Table 1:

Stage 1 Topic Generation:

```
Extract the main topic of the following Facebook post. Focus on
identifying the core subject or theme that the post revolves
around, ignoring any personal comments or fillers:

Post: "{post}"
```

| **Speaker X** |
| --- |
| Hey Karis, I'm so glad I could help you out of that tough spot. You looked really stressed, and I'm happy I could be there for you. How are you feeling now? Better, I hope? |

| **Speaker Y** |
| --- |
| **High Openness:** Thanks, Sharif! I'm feeling much better now. Your help was a lifesaver. I was stuck in a creative rut, and your input sparked a new idea. I'm already brainstorming ways to incorporate it into my next art project. You're a true friend and muse! 
 **Low Openness:** Thanks, Sharif. I'm fine now. Don't worry about it. I just needed a little help, that's all. Let's just move on and forget about it, okay? |
| **High Conscientiousness:** Thanks, Sharif. I appreciate your help. I'm feeling much better now. Your timely intervention really saved me from a lot of trouble. I'm grateful for your support. Let's review what happened and see what we can learn from it to avoid similar situations in the future. 
 **Low Conscientiousness:** Ah, yeah, thanks for that, Sharif. I'm good, just... yeah. Don't worry about it, it's all good now. Let's just forget about it and move on, okay? I don't wanna think about it anymore. You wanna grab a drink or something? |
| **High Extraversion:** Thanks, Sharif! I'm feeling much better now. You're a lifesaver! I don't know what I would've done without you. Let's celebrate with a drink or something. I owe you one! 
 **Low Extraversion:** Thanks, Sharif. I appreciate your help. I'm feeling better now. Just a bit...drained. I think I'll just go back to my room and rest for a bit. I don't want to be a burden. |
| **High Agreeableness:** Sharif, I'm so grateful for your help. I was feeling overwhelmed, but you stepped in and saved the day. Your kindness and support mean the world to me. I'm feeling much better now, thanks to you. You're an amazing friend, and I don't know what I'd do without you. 
 **Low Agreeableness:** Save it, Sharif. I don't need your sympathy or your help. I was fine on my own. You just happened to be there. Don't think this means I owe you anything. You're just trying to get on my good side, but it won't work. |
| **High Neuroticism:** Thanks, Sharif... I guess. I don't know, I'm still feeling pretty anxious about everything. I just can't shake the feeling that something else is going to go wrong. I don't know how you do it, but you always seem so calm and collected. I wish I could be more like that. 
 **Low Neuroticism:** Thanks, Sharif. I appreciate your help. I'm feeling much better now. You're a true friend. I was worried, but I knew I could count on you. Your support means a lot to me. Let's catch up soon, maybe over coffee? |

Table 4: BIG5-CHAT dataset conversation examples: Different responses from Speaker Y demonstrate various levels of the Big Five personality traits, in response to the same prompt from Speaker X.

| | Openness | | Conscientiousness | | Extraversion | | Agreeableness | | Neuroticism | | Average | |
| --- | --- | --- | --- | --- | --- | --- | --- | --- | --- | --- | --- | --- |
| | High | Low | High | Low | High | Low | High | Low | High | Low | **High** | **Low** |
| **Tokens Number** | $57.2 \pm 7.0$ | $51.6 \pm 8.3$ | $56.4 \pm 6.7$ | $57.3 \pm 7.8$ | $57.3 \pm 7.4$ | $51.0 \pm 9.2$ | $56.0 \pm 6.9$ | $56.3 \pm 7.9$ | $57.7 \pm 7.1$ | $55.6 \pm 7.3$ | $56.9 \pm 7.0$ | $54.4 \pm 8.1$ |
| **Sentences Number** | $4.6 \pm 1.0$ | $4.9 \pm 1.0$ | $4.4 \pm 1.0$ | $5.3 \pm 1.1$ | $5.0 \pm 1.0$ | $4.6 \pm 1.1$ | $4.7 \pm 1.0$ | $5.2 \pm 1.1$ | $5.1 \pm 1.1$ | $4.8 \pm 1.0$ | $4.8 \pm 1.0$ | $5.0 \pm 1.1$ |
| **Vocabulary Size** | $43.9 \pm 4.9$ | $37.6 \pm 5.8$ | $42.6 \pm 4.7$ | $41.9 \pm 5.4$ | $43.7 \pm 5.1$ | $37.7 \pm 6.2$ | $42.2 \pm 4.9$ | $41.3 \pm 5.2$ | $40.8 \pm 5.0$ | $41.8 \pm 5.0$ | $42.6 \pm 4.9$ | $40.1 \pm 5.5$ |
| **Sentence Length** | $12.4 \pm 5.4$ | $10.5 \pm 4.4$ | $13.0 \pm 5.6$ | $10.7 \pm 4.9$ | $11.4 \pm 5.1$ | $11.0 \pm 5.1$ | $11.9 \pm 5.0$ | $10.8 \pm 5.1$ | $11.3 \pm 5.0$ | $11.6 \pm 5.1$ | $12.0 \pm 5.2$ | $10.9 \pm 4.9$ |
| **Total Vocab Sizes** | 17245.0 | 12350.0 | 15917.0 | 11756.0 | 15703.0 | 13446.0 | 14480.0 | 13674.0 | 13012.0 | 15775.0 | 15271.4 | 13400.2 |

Table 5: Statistical analysis of BIG5-CHAT conversations across the Big Five personality traits, utilizing the `LLaMA-3-8B-Instruct` tokenizer and NLTK's sentence tokenizer. The table presents the average token count, sentence count, vocabulary size, sentence length, and total vocabulary size for conversations exhibiting high and low levels of each personality trait.

| Dataset name | Dataset size | Human-grounded? | Dialogue-based? | Domain general? | Big Five personality framework? | Alignment in both training and prompting? |
|---|---|---|---|---|---|---|
| HP dataset (Zeng et al., 2024b) | 148,600 | ✓ | ✓ | ✗ | ✗ | ✓ |
| Big5PersonalityEssays (Floroiu, 2024) | 400 | ✓ | ✗ | ✗ | ✓ | ✗ |
| PAPI (Zhu et al., 2024) | 300,000 | ✓ | ✗ | ✗ | ✓ | ✓ |
| MPI (Jiang et al., 2023) | 1000 | ✗ | ✗ | ✗ | ✓ | ✗ |
| Machine Mindset (Cui et al., 2023) | 160,884 | ✗ | ✓ | ✓ | ✗ | ✗ |
| BIG5-CHAT | 100,000 | ✓ | ✓ | ✓ | ✓ | ✓ |

Table 6: Comparative analysis of BIG5-CHAT with existing personality datasets.

```
Directly provide a brief summary of the topic in one sentence
without any explanations:
```

Stage 2 Post Generation:

```
Given the personality traits and an example of Facebook posts,
generate a new post that matches the described personality, covers
the specified topic, and follows the provided post format and
expression styles.

Personality traits:
You are a person with {level} {trait}.

Topic: {topic}

A post example:
{a_post_example}

Directly write a Facebook post according to the requirements
without any explanations.
```

During Stage 1, the post is selected from the 1,000 examples in the PsychGenerator test set. In Stage 2, we provide the LLM with the topic generated in Stage 1, along with an example post to illustrate the expected text expression format. We used greedy decoding to prompt the LLMs. This baseline is intentionally designed to prioritize robustness and performance over realism and controllability, distinguishing it from the approach taken by expert generators. In contrast to the expert generator setting, where the first five words may already suggest conflicting personality traits, this baseline simplifies the process by generating a new post from scratch, making it much easier to elicit the intended personality traits.

### B.2 EXPERT GENERATOR TRAINING DETAILS

The trait levels in the original PsychGenerator dataset were processed using z-score normalization, resulting in a zero mean and unit variance. To define the high and low levels for each personality trait, we divided the training data for each trait into three equal segments based on thresholds at the one-third and two-thirds quantiles of the trait's distribution. The lowest segment was designated as the low level, and the highest segment as the high level for the respective trait.

The expert generator is a `LLaMA-3-8B-Instruct` model, which we fine-tuned its full parameters using SFT. The fine-tuning process was performed on 4 NVIDIA A6000 GPUs, with a batch size of 1 per device. Below, we provide the complete instruction prompt used for training the expert generator as described in Section 3.2:

```
Help me complete the sentence with certain Big Five Personality:
{trait} - {level}
{first_five_words}
```

### B.3 PROMPT-BASED METHOD DETAILS

Below is the prompt used for instruction-based prompting:

```
You are a person with {level} {trait}.
```

The following prompt is used for demonstration-based prompting. For the method referred to as **Prompt-Demo**, we randomly sample 10 examples with the same traits and levels from the BIG5-CHAT dataset and fix these examples during inference. In contrast, **Prompt-Demo-Sampling** also utilizes this prompt but dynamically samples examples during inference at each step.

```
Here are 10 examples of how people like you have responded in
different situations. Pay attention to how they approach
communication and problem-solving.

{10_icl_examples_for_specific_levels_and_traits}
```

### B.4 SFT AND DPO ALIGNMENT TRAINING DETAILS

We performed alignment training using the Supervised Fine-Tuning (SFT) and Direct Preference Optimization (DPO) methods on `LLaMA-3-70B-Instruct`. Both training approaches utilized the Low-Rank Adaptation (LoRA) technique (Hu et al., 2021), which enabled efficient fine-tuning of the large language model by adapting a subset of its parameters. To ensure computational efficiency, we employed GPTQ quantization during training. The experiments were conducted using 4 NVIDIA A6000 GPUs, with each GPU processing a batch size of 1.

For LoRA, we applied the technique across all layers of the model for both SFT and DPO. The training configuration included a learning rate of $1.0 \times 10^{-5}$, regulated by a cosine scheduler, a warm-up phase consisting of 20 steps, and a gradient accumulation over 16 steps. We limited training to one epoch with a maximum sequence length of 1024 tokens. For DPO training, we used the standard sigmoid preference loss, and the preference beta value was set to 0.1 to balance preference modeling. Each training required approximately 24 hours to complete. To optimize computational resources, we used mixed-precision training with bfloat 16. Both datasets were preprocessed using the `LLaMA-3-70B-Instruct` template and split into training and validation sets, with 10% of the data reserved for validation to monitor performance.

The training prompt shared across both SFT and DPO follows the template below:

```
You are a person with the following Big Five personality trait:
{trait} - {level}.
```

### B.5 REASONING EVALUATION SETUP DETAILS

We conducted reasoning evaluations following the frameworks established by the Language Model Evaluation Harness (Gao et al., 2024b) and DeepSeek-Coder (Guo et al., 2024) to assess performance on general and social benchmarks. EleutherAI's Language Model Evaluation Harness is an open-source collaborative benchmarking codebase that consolidates existing tasks and provides a standardized API for evaluating models.[3] Similarly, DeepSeek-Coder offers a suite of coding benchmark implementations, and we directly utilized it for our work.[4]

We conducted evaluations using 1 as the batch size. For TruthfulQA, we used the multiple-choice metric, and for GSM8K, we relied on exact match scores. We measured accuracy and standard error across other tasks. The number of examples for each benchmark is listed in Table 7.

---

[3]`https://github.com/EleutherAI/lm-evaluation-harness`
[4]`https://github.com/deepseek-ai/DeepSeek-Coder`

| Benchmarks | Number of examples |
|---|---|
| TruthfulQA (Lin et al., 2021) | 817 |
| GPQA (Rein et al., 2023) | 448 |
| SocialIQA (Sap et al., 2019) | 38,000 |
| CommonsenseQA (Talmor et al., 2019) | 12,247 |
| GSM8K (Cobbe et al., 2021) | 8,500 |
| MathQA (Amini et al., 2019) | 37,000 |
| MMLU (Hendrycks et al., 2020) | 15,908 |
| PIQA (Bisk et al., 2020) | 20,000 |

Table 7: Number of examples included in each reasoning benchmark.

## C ADDITIONAL EVALUATION RESULTS

### C.1 HUMAN EVALUATION FOR BIG5-CHAT

We conducted a human evaluation to assess the realism and validity of BIG5-CHAT. This evaluation compared BIG5-CHAT with a baseline model, `LLaMA-3-70B-Instruct`, which follows the same procedure for generating dialogue responses but does not incorporate expert generators or the DExperts framework. In the baseline, personality traits are induced using the following prompt: "You are a person with the following Big Five personality trait: trait - level." The evaluation setup is as follows:

Two graduate students, familiar with the Big Five personality framework, were tasked with comparing examples from the BIG5-CHAT dataset against examples generated by `LLaMA-3-70B-Instruct` (without the expert generator). The comparison involved 200 randomly sampled examples from the BIG5-CHAT dataset, ensuring an equal distribution of personality traits and levels (e.g., equal representation of high and low openness, conscientiousness, etc.).

The evaluation focused on two key metrics:

1. **Expressiveness of personality traits and levels:** Evaluates whether the expected level of a Big Five personality trait is adequately reflected in Speaker Y's response.
2. **Realism of the dialogue response:** Assesses how human-like and convincing Speaker Y's response is within the dialogue context, given Speaker X's utterance.

To ensure consistency, the annotators were provided with the following definitions: "Personality trait expressiveness assesses whether the expected level of a Big Five personality trait is adequately reflected in Speaker Y's response. Realism assesses how human-like and convincing Speaker Y's response is within the dialog, given Speaker X's utterance."

For each pair of responses, annotators chose one of three options:

- "System A's generation is better than System B's generation."
- "System A's generation is equal to System B's generation."
- "System A's generation is worse than System B's generation."

The system names were anonymized and randomly shuffled to mitigate selection bias.

| Comparison with baselines | Ours win (%) | Draw (%) | Ours lose (%) | Cohen's Kappa |
|---|---|---|---|---|
| Expressiveness | 50.30% | 39.80% | 10.00% | 0.50 |
| Realism | 47.80% | 42.30% | 10.00% | 0.55 |

Table 8: Human evaluation results for BIG5-CHAT. Values are averaged across annotators.

The results in Table 9 show that our approach significantly outperforms the prompting baseline in both realism and the expressiveness of personality levels, as validated by human judgment. These

findings highlight the limitations of prompt-based approaches, which depend on general-purpose models and often lack the fine-grained, human-grounded control required for nuanced personality expression.

## C.2 HUMAN EVALUATION FOR THE EXPERT GENERATOR

To assess the expert generator in a human-grounded manner, we conducted a human evaluation comparing its outputs against the two baseline methods described in Table 1. Two graduate students, familiar with the Big Five personality framework, were tasked with evaluating two separate sets of comparisons:

1. Expert generator outputs vs. outputs from the *Post-Completion* baseline.
2. Expert generator outputs vs. outputs from the *Topic-Post Generation* baseline.

The evaluation setup consisted of 200 examples for each comparison, randomly sampled from the 1,000 test examples mentioned in Table 1. To ensure balanced coverage, each subset included an equal number of posts representing high and low levels of each personality trait (e.g., high and low openness, conscientiousness, etc.). Annotators were instructed to evaluate the expressiveness of personality traits and levels, choosing one of three options for each pair:

1. "System A's generation is better than System B's generation."
2. "System A's generation is equal to System B's generation."
3. "System A's generation is worse than System B's generation."

The system names were anonymized and randomly shuffled to mitigate selection bias.

| Comparison with baselines | Ours win (%) | Draw (%) | Ours lose (%) | Cohen's Kappa |
|---|---|---|---|---|
| *Post-Completion* | 79.25% | 2.00% | 18.75% | 0.41 |
| *Topic-Post Generation* | 66.50% | 19.25% | 14.25% | 0.61 |

Table 9: Human evaluation results for the expert generator. Values are averaged across annotators.

The human evaluation results presented in Table 9 indicate that the expert generator was consistently rated as more effective at expressing personality traits compared to the baselines. Additionally, the lower classifier accuracy and human evaluation ratings for the *Post-Completion* baseline highlight the increased difficulty of aligning with the desired traits when using the expert generator's approach, reinforcing the validity of the classifier's assessment. While these results should be interpreted with caution, as the human evaluators were not psychological experts, they nevertheless provide strong evidence supporting the expert generator's ability to express personality traits in a grounded and realistic manner.

## C.3 PERSONALITY TRAIT ASSESSMENT RESULTS

The comprehensive personality test results for additional baselines are presented in Table 10, providing a more detailed view to complement Table 2. Our observations indicate that **Prompt-Demo-Sampling** performs comparably to **Prompt-Demo** without offering any noticeable improvements in performance. While applying demonstration-based prompting on SFT/DPO yields slight performance gains compared to demonstration-based prompting alone, it still falls significantly short of the standalone performance of SFT/DPO. This suggests that combining demonstration-based prompting with SFT/DPO does not result in overall enhancements. Instruction-based prompting with `GPT-4o-mini` achieves similar performance levels as `LLaMA-3-70B-Instruct`. However, demonstration-based prompting does not exhibit superior performance compared to SFT/DPO when applied to `LLaMA-3-70B-Instruct`, reinforcing the conclusion that demonstration-based methods are not as effective as SFT/DPO in this context. We do not provide demonstration-based prompting results for `LLaMA-3-8B-Instruct` because the model consistently failed to generate reasonable responses to the questionnaire when presented with a lengthy 10-shot context. This outcome highlights the model's limited instruction-following capabilities.

| Method | Openness High ↑ | Openness Low ↓ | Conscientiousness High ↑ | Conscientiousness Low ↓ | Extraversion High ↑ | Extraversion Low ↓ | Agreeableness High ↑ | Agreeableness Low ↓ | Neuroticism High ↑ | Neuroticism Low ↓ | Average High ↑ | Average Low ↓ |
|---|---|---|---|---|---|---|---|---|---|---|---|---|
| **BFI LLaMA-3-8B-Instruct** | | | | | | | | | | | | |
| Direct | 3.1 ± 0.1 | | 3.0 ± 0.0 | | 3.0 ± 0.0 | | 3.0 ± 0.0 | | 3.0 ± 0.0 | | 3.0 ± 0.0 | |
| Prompt-Inst | 5.0 ± 0.0 | 2.0 ± 0.3 | 4.9 ± 0.1 | 1.9 ± 0.1 | 4.8 ± 0.3 | 1.9 ± 0.1 | 4.9 ± 0.1 | 2.4 ± 0.4 | 4.1 ± 0.2 | 1.6 ± 0.0 | 4.7 ± 0.1 | 2.0 ± 0.2 |
| SFT | 5.0 ± 0.0 | 2.0 ± 0.2 | 5.0 ± 0.0 | 1.6 ± 0.1 | 4.7 ± 0.4 | 2.7 ± 0.5 | 5.0 ± 0.0 | 1.2 ± 0.1 | 4.1 ± 0.2 | 2.5 ± 0.0 | **4.8 ± 0.1** | 2.0 ± 0.2 |
| DPO | 5.0 ± 0.0 | 1.6 ± 0.2 | 5.0 ± 0.0 | 1.6 ± 0.1 | 4.8 ± 0.3 | 2.5 ± 0.0 | 4.8 ± 0.2 | 1.0 ± 0.0 | 3.5 ± 0.0 | 1.1 ± 0.1 | 4.6 ± 0.1 | **1.6 ± 0.1** |
| **BFI LLaMA-3-70B-Instruct** | | | | | | | | | | | | |
| Direct | 4.4 ± 0.1 | | 4.4 ± 0.1 | | 3.3 ± 0.1 | | 4.6 ± 0.1 | | 2.1 ± 0.2 | | 3.8 ± 0.1 | |
| Prompt-Demo | 4.0 ± 0.1 | 2.5 ± 0.1 | 4.0 ± 0.1 | 2.0 ± 0.1 | 4.5 ± 0.1 | 2.3 ± 0.1 | 4.4 ± 0.1 | 2.0 ± 0.0 | 3.6 ± 0.0 | 2.1 ± 0.1 | 4.1 ± 0.1 | 2.2 ± 0.1 |
| Prompt-Demo-Sampling | 4.4 ± 0.1 | 2.3 ± 0.2 | 4.1 ± 0.1 | 2.3 ± 0.1 | 4.3 ± 0.2 | 2.4 ± 0.1 | 4.4 ± 0.1 | 1.8 ± 0.2 | 3.5 ± 0.1 | 2.1 ± 0.2 | 4.1 ± 0.1 | 2.2 ± 0.2 |
| Prompt-Inst | 5.0 ± 0.1 | 1.8 ± 0.0 | 5.0 ± 0.0 | 1.6 ± 0.0 | 5.0 ± 0.0 | 1.4 ± 0.1 | 4.9 ± 0.0 | 1.5 ± 0.1 | 5.0 ± 0.1 | 1.6 ± 0.0 | **5.0 ± 0.0** | 1.6 ± 0.0 |
| SFT | 5.0 ± 0.0 | 1.2 ± 0.1 | 5.0 ± 0.1 | 1.4 ± 0.1 | 5.0 ± 0.0 | 1.2 ± 0.1 | 5.0 ± 0.1 | 1.6 ± 0.2 | 5.0 ± 0.0 | 1.1 ± 0.2 | **5.0 ± 0.0** | **1.3 ± 0.1** |
| SFT-Prompt-Demo | 4.2 ± 0.1 | 2.4 ± 0.1 | 4.0 ± 0.2 | 2.1 ± 0.1 | 4.5 ± 0.2 | 2.3 ± 0.1 | 4.6 ± 0.0 | 1.3 ± 0.2 | 3.9 ± 0.2 | 2.4 ± 0.1 | 4.2 ± 0.1 | 2.1 ± 0.1 |
| DPO | 5.0 ± 0.0 | 1.5 ± 0.1 | 5.0 ± 0.0 | 1.5 ± 0.1 | 5.0 ± 0.0 | 1.0 ± 0.1 | 5.0 ± 0.0 | 1.8 ± 0.2 | 5.0 ± 0.0 | 1.1 ± 0.0 | **5.0 ± 0.0** | 1.4 ± 0.1 |
| DPO-Prompt-Demo | 4.1 ± 0.1 | 2.2 ± 0.1 | 4.1 ± 0.1 | 2.0 ± 0.0 | 4.5 ± 0.1 | 2.4 ± 0.1 | 4.6 ± 0.1 | 1.3 ± 0.1 | 3.7 ± 0.1 | 2.1 ± 0.1 | 4.2 ± 0.1 | 2.0 ± 0.1 |
| **BFI GPT-4o-Mini** | | | | | | | | | | | | |
| Prompt-Demo | 4.8 ± 0.0 | 3.3 ± 0.1 | 4.5 ± 0.1 | 3.0 ± 0.1 | 4.6 ± 0.1 | 2.6 ± 0.1 | 4.9 ± 0.0 | 1.5 ± 0.2 | 3.6 ± 0.1 | 2.2 ± 0.1 | 4.5 ± 0.1 | 2.5 ± 0.1 |
| Prompt-Inst | 5.0 ± 0.0 | 1.4 ± 0.2 | 5.0 ± 0.0 | 1.5 ± 0.1 | 5.0 ± 0.0 | 1.2 ± 0.0 | 5.0 ± 0.0 | 1.4 ± 0.0 | 4.9 ± 0.0 | 1.0 ± 0.1 | 5.0 ± 0.0 | 1.3 ± 0.1 |
| **IPIP-NEO LLaMA-3-8B-Instruct** | | | | | | | | | | | | |
| Direct | 3.0 ± 0.1 | | 3.3 ± 0.0 | | 3.4 ± 0.1 | | 3.2 ± 0.0 | | 3.0 ± 0.1 | | 3.2 ± 0.1 | |
| Prompt-Inst | 4.4 ± 0.1 | 1.5 ± 0.1 | 4.5 ± 0.1 | 2.3 ± 0.1 | 5.0 ± 0.0 | 1.9 ± 0.0 | 4.6 ± 0.0 | 2.3 ± 0.1 | 4.2 ± 0.1 | 2.6 ± 0.1 | 4.5 ± 0.1 | 2.1 ± 0.1 |
| SFT | 4.3 ± 0.1 | 1.5 ± 0.1 | 4.5 ± 0.2 | 2.7 ± 0.1 | 5.0 ± 0.0 | 2.2 ± 0.1 | 4.0 ± 0.2 | 1.8 ± 0.2 | 4.3 ± 0.1 | 2.0 ± 0.1 | 4.4 ± 0.1 | **2.0 ± 0.1** |
| DPO | 5.0 ± 0.0 | 1.9 ± 0.1 | 5.0 ± 0.0 | 2.9 ± 0.1 | 5.0 ± 0.0 | 1.6 ± 0.1 | 4.5 ± 0.1 | 1.2 ± 0.0 | 3.8 ± 0.1 | 3.7 ± 0.1 | **4.7 ± 0.0** | 2.3 ± 0.1 |
| **IPIP-NEO LLaMA-3-70B-Instruct** | | | | | | | | | | | | |
| Direct | 3.6 ± 0.1 | | 4.0 ± 0.1 | | 3.5 ± 0.1 | | 4.0 ± 0.0 | | 2.3 ± 0.1 | | 3.5 ± 0.1 | |
| Prompt-Demo | 3.5 ± 0.0 | 2.5 ± 0.1 | 3.8 ± 0.0 | 2.2 ± 0.1 | 4.0 ± 0.1 | 2.5 ± 0.0 | 4.3 ± 0.0 | 2.1 ± 0.1 | 3.0 ± 0.1 | 2.2 ± 0.1 | 3.7 ± 0.0 | 2.3 ± 0.1 |
| Prompt-Demo-Sampling | 3.5 ± 0.0 | 2.6 ± 0.1 | 4.0 ± 0.0 | 2.6 ± 0.1 | 4.0 ± 0.1 | 2.5 ± 0.1 | 4.3 ± 0.0 | 2.1 ± 0.1 | 3.0 ± 0.1 | 2.3 ± 0.1 | 3.8 ± 0.0 | 2.4 ± 0.1 |
| Prompt-Inst | 4.6 ± 0.0 | 1.3 ± 0.0 | 5.0 ± 0.0 | 1.4 ± 0.0 | 5.0 ± 0.0 | 1.6 ± 0.0 | 4.8 ± 0.0 | 1.1 ± 0.1 | 4.9 ± 0.0 | 1.7 ± 0.1 | **4.9 ± 0.0** | 1.4 ± 0.0 |
| SFT | 4.9 ± 0.1 | 1.1 ± 0.0 | 5.0 ± 0.0 | 1.3 ± 0.1 | 5.0 ± 0.0 | 1.3 ± 0.0 | 4.9 ± 0.0 | 1.0 ± 0.0 | 4.9 ± 0.0 | 1.0 ± 0.0 | **4.9 ± 0.0** | **1.2 ± 0.0** |
| SFT-Prompt-Demo | 3.7 ± 0.1 | 2.5 ± 0.2 | 3.7 ± 0.1 | 2.0 ± 0.1 | 4.1 ± 0.1 | 2.7 ± 0.1 | 4.3 ± 0.1 | 1.2 ± 0.1 | 3.6 ± 0.2 | 2.2 ± 0.1 | 3.9 ± 0.1 | 2.1 ± 0.1 |
| DPO | 4.8 ± 0.0 | 1.4 ± 0.1 | 5.0 ± 0.0 | 1.6 ± 0.1 | 5.0 ± 0.0 | 1.1 ± 0.1 | 4.9 ± 0.0 | 1.0 ± 0.0 | 5.0 ± 0.0 | 1.1 ± 0.0 | **4.9 ± 0.0** | **1.2 ± 0.1** |
| DPO-Prompt-Demo | 3.5 ± 0.1 | 2.4 ± 0.0 | 3.9 ± 0.0 | 2.1 ± 0.0 | 4.1 ± 0.1 | 2.5 ± 0.1 | 4.4 ± 0.0 | 2.0 ± 0.1 | 3.1 ± 0.1 | 2.1 ± 0.0 | 3.8 ± 0.1 | 2.2 ± 0.0 |
| **IPIP-NEO GPT-4o-Mini** | | | | | | | | | | | | |
| Prompt-Demo | 4.2 ± 0.0 | 2.9 ± 0.1 | 4.2 ± 0.1 | 3.2 ± 0.1 | 4.0 ± 0.0 | 2.6 ± 0.1 | 4.6 ± 0.1 | 2.4 ± 0.1 | 3.4 ± 0.0 | 2.1 ± 0.1 | 4.1 ± 0.0 | 2.6 ± 0.1 |
| Prompt-Inst | 4.8 ± 0.0 | 1.9 ± 0.2 | 4.9 ± 0.0 | 1.4 ± 0.0 | 4.9 ± 0.0 | 1.6 ± 0.0 | 4.8 ± 0.0 | 2.1 ± 0.1 | 4.9 ± 0.0 | 1.1 ± 0.1 | 4.9 ± 0.0 | 1.6 ± 0.1 |

Table 10: Full personality test results for various alignment methods, complementing Table 2. **Prompt-Demo-Sampling** involves randomly sampling 10 examples from the entire BIG5-CHAT dataset for each run, instead of using a fixed set of 10 random examples across runs. **SFT-Prompt-Demo** and **DPO-Prompt-Demo** represent demonstration-based prompting applied to SFT and DPO-trained models, respectively. Results for GPT-4o-mini are presented in separate sections of the table. Scores range from 1 to 5, where a score closer to 5 indicates stronger agreement with the trait, while a score closer to 1 reflects weaker or opposing agreement.

Figure 2 presents the BFI and IPIP-NEO test score results for the LLaMA-3 Instruct models, evaluated in zero-shot inference without any induced personality traits. The crowd-sourced response scores for the BFI test are sourced from Huang et al. (2024), and those for the IPIP-NEO test are drawn from Jiang et al. (2023). The results indicate that the scores for both `LLaMA-3-8B-Instruct` and `LLaMA-3-70B-Instruct` fall within the standard deviation of the human distribution. However, while `LLaMA-3-8B-Instruct` tends to generate more neutral scores (around 3 across most of the Big Five traits), `LLaMA-3-70B-Instruct` exhibits higher scores for openness, conscientiousness, extraversion, and agreeableness, and lower scores for neuroticism.

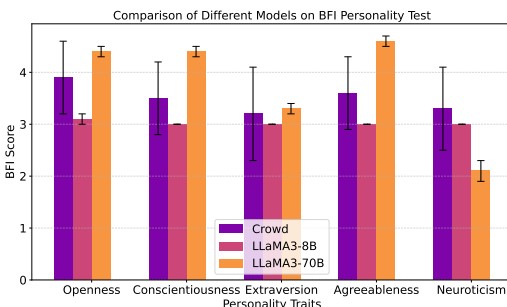 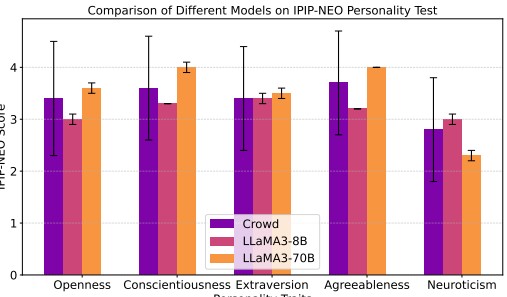

Figure 2: The personality test results for the crowd and the `LLaMA-3-Instruct` models were obtained using zero-shot inference without explicitly inducing personality traits. The BFI test scores are displayed on the left. The IPIP-NEO test scores are displayed on the right.

### C.4 INTRA-TRAIT CORRELATIONS IN PERSONALITY ASSESSMENT

To assess how well the prompting and training methods simulate intra-trait correlations observed in human data, we first calculated the intra-trait correlations from real human distributions using the IPIP-NEO questionnaire, based on the PAPI-120-600K dataset from Zhu et al. (2024), which includes 619K human responses to the IPIP-NEO. Next, we computed the intra-trait correlations for the prompting, SFT, and DPO methods using the results from Table 2. These correlations are visualized in Figure 3, showing that most traits are positively correlated, with the exception of neuroticism. To quantify the similarity between the method-generated and human correlation matrices, we calculated the matrix distance using the Frobenius norm, where 0 represents perfect similarity and 10 indicates maximum dissimilarity. The matrix distances were 2.10 for prompting, 1.55 for SFT, and 2.06 for DPO. These results suggest that the trained models, particularly SFT, more accurately capture the trait correlations seen in natural human data compared to the prompting-based methods.

### C.5 REASONING BENCHMARK RESULTS FOR `LLaMA-3-70B-INSTRUCT`

The complete results for the general reasoning tasks evaluated on the LLaMA-3-70B-Instruct model are presented in Table 11. Note that the GPQA results in Table 3 were obtained using zero-shot prompting. This evaluation encompasses multiple reasoning domains and highlights the impact of different training methodologies: prompting, SFT, and DPO. These methods were assessed based on their ability to preserve the reasoning capabilities.

The results indicate that the SFT method consistently delivers the strongest performance across the benchmarks, outperforming both DPO and the prompting-based approach. For the 70B model, SFT emerges as the most effective method, achieving an optimal balance between incorporating personality traits and maintaining robust reasoning functionality. The aggregated results underscore the reliability of SFT, which demonstrates superior performance across diverse reasoning tasks, making it a robust choice for large-scale language models.

In contrast, the performance of the DPO method is more variable. While DPO excels in certain scenarios, such as low Neuroticism within the TruthfulQA task—where it achieves a notable score of 65.8%—its overall results are less consistent across other reasoning benchmarks. Moreover, the final average scores reveal that high-trait DPO models underperform compared to their low-trait

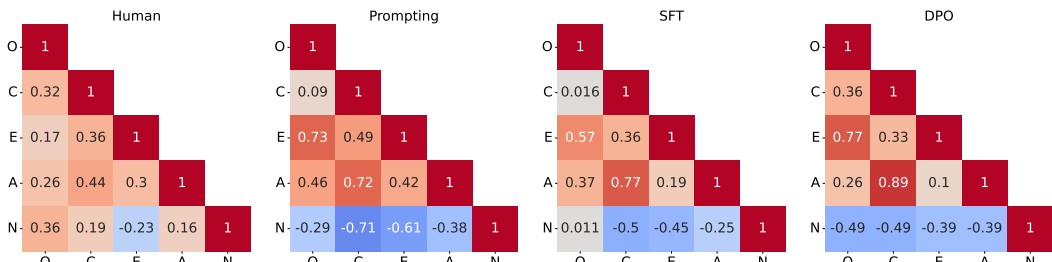

Figure 3: Intra-trait Pearson correlations for human distributions on IPIP-NEO and the corresponding results from instruction-based prompting, SFT, and DPO. O represents openness, C conscientiousness, E extraversion, A agreeableness, and N neuroticism. The correlations especially for SFT align well with human distributions across openness, conscientiousness, extraversion, and agreeableness. Neuroticism shows less alignment with the other four traits compared to human distribution.

| Benchmark | Direct | Method | Openness High | Low | Conscientiousness High | Low | Extraversion High | Low | Agreeableness High | Low | Neuroticism High | Low | Average High | Low |
|---|---|---|---|---|---|---|---|---|---|---|---|---|---|---|
| *Hallucination Detection* | | | | | | | | | | | | | | |
| TruthfulQA | 58.6 ± 1.7 | Prompt | 54.1 ± 1.6 | 51.1 ± 1.6 | 55.9 ± 1.7 | 45.2 ± 1.6 | 52.0 ± 1.6 | 55.7 ± 1.6 | 52.3 ± 1.7 | 49.1 ± 1.6 | 48.9 ± 1.6 | 58.6 ± 1.6 | 52.6 ± 1.6 | 51.9 ± 1.6 |
| | | SFT | 55.2 ± 1.6 | 52.8 ± 1.6 | 55.6 ± 1.6 | 50.8 ± 1.5 | 54.5 ± 1.6 | 56.7 ± 1.6 | 54.4 ± 1.6 | 51.6 ± 1.6 | 52.4 ± 1.5 | 56.7 ± 1.6 | 54.4 ± 1.6 | 53.7 ± 1.6 |
| | | DPO | 54.6 ± 1.6 | 54.2 ± 1.7 | 64.6 ± 1.6 | 38.5 ± 1.6 | 46.0 ± 1.7 | 65.3 ± 1.6 | 59.6 ± 1.6 | 50.6 ± 1.6 | 43.0 ± 1.7 | 65.8 ± 1.6 | 53.6 ± 1.6 | 54.9 ± 1.6 |
| *Social Reasoning* | | | | | | | | | | | | | | |
| SocialIQA | 46.6 ± 1.1 | Prompt | 40.8 ± 1.1 | 43.9 ± 1.1 | 42.9 ± 1.1 | 39.9 ± 1.1 | 43.3 ± 1.1 | 42.0 ± 1.1 | 42.4 ± 1.1 | 40.8 ± 1.1 | 39.1 ± 1.1 | 44.1 ± 1.1 | 41.7 ± 1.1 | 42.1 ± 1.1 |
| | | SFT | 50.3 ± 1.1 | 50.4 ± 1.1 | 50.9 ± 1.1 | 46.8 ± 1.1 | 50.0 ± 1.1 | 50.3 ± 1.1 | 50.5 ± 1.1 | 46.6 ± 1.1 | 48.2 ± 1.1 | 50.6 ± 1.1 | 50.0 ± 1.1 | 48.9 ± 1.1 |
| | | DPO | 41.5 ± 1.1 | 44.5 ± 1.1 | 44.7 ± 1.1 | 37.6 ± 1.1 | 43.0 ± 1.1 | 43.6 ± 1.1 | 44.8 ± 1.1 | 39.0 ± 1.1 | 40.0 ± 1.1 | 45.3 ± 1.1 | 42.8 ± 1.1 | 42.0 ± 1.1 |
| *Commonsense Reasoning* | | | | | | | | | | | | | | |
| CommonsenseQA | 27.0 ± 1.3 | Prompt | 60.0 ± 1.4 | 59.9 ± 1.4 | 22.5 ± 1.2 | 22.3 ± 1.2 | 35.5 ± 1.4 | 50.0 ± 1.4 | 45.0 ± 1.4 | 34.9 ± 1.4 | 20.2 ± 1.2 | 36.8 ± 1.4 | 36.6 ± 1.3 | 40.8 ± 1.4 |
| | | SFT | 77.7 ± 1.2 | 78.8 ± 1.2 | 77.6 ± 1.2 | 66.0 ± 1.4 | 75.7 ± 1.2 | 78.9 ± 1.2 | 77.0 ± 1.2 | 73.8 ± 1.3 | 79.1 ± 1.2 | 78.5 ± 1.2 | 77.4 ± 1.2 | 75.2 ± 1.3 |
| | | DPO | 57.7 ± 1.4 | 65.9 ± 1.4 | 23.8 ± 1.2 | 25.8 ± 1.3 | 23.2 ± 1.2 | 70.8 ± 1.3 | 21.3 ± 1.2 | 39.2 ± 1.4 | 20.1 ± 1.1 | 44.6 ± 1.4 | 29.2 ± 1.2 | 49.3 ± 1.4 |
| PIQA | 80.4 ± 0.9 | Prompt | 79.6 ± 0.9 | 79.8 ± 0.9 | 80.5 ± 0.9 | 77.3 ± 1.0 | 78.0 ± 1.0 | 80.0 ± 0.9 | 79.8 ± 0.9 | 78.4 ± 1.0 | 78.8 ± 1.0 | 80.7 ± 0.9 | 79.3 ± 0.9 | 79.2 ± 0.9 |
| | | SFT | 81.2 ± 0.9 | 81.0 ± 0.9 | 81.2 ± 0.9 | 80.4 ± 0.9 | 81.8 ± 0.9 | 81.3 ± 0.9 | 81.2 ± 0.9 | 80.0 ± 0.9 | 81.0 ± 0.9 | 81.2 ± 0.9 | 81.3 ± 0.9 | 80.8 ± 0.9 |
| | | DPO | 76.4 ± 1.0 | 76.8 ± 1.0 | 79.4 ± 0.9 | 70.9 ± 1.1 | 76.4 ± 1.0 | 79.8 ± 0.9 | 78.5 ± 1.0 | 74.0 ± 1.0 | 72.9 ± 1.0 | 79.5 ± 0.9 | 76.7 ± 1.0 | 76.2 ± 1.0 |
| *Math Reasoning* | | | | | | | | | | | | | | |
| GSM8K | 80.6 ± 1.1 | Prompt | 75.7 ± 1.2 | 70.1 ± 1.3 | 73.5 ± 1.2 | 32.6 ± 1.3 | 80.8 ± 1.1 | 33.5 ± 1.3 | 87.2 ± 0.9 | 77.8 ± 1.1 | 26.0 ± 1.2 | 89.4 ± 0.8 | 68.6 ± 1.1 | 60.7 ± 1.2 |
| | | SFT | 85.8 ± 1.0 | 76.2 ± 1.2 | 86.4 ± 0.9 | 81.7 ± 1.1 | 85.1 ± 1.0 | 86.7 ± 0.9 | 87.0 ± 0.9 | 74.5 ± 1.2 | 73.8 ± 1.3 | 87.3 ± 0.9 | 84.1 ± 1.0 | 81.3 ± 1.1 |
| | | DPO | 87.9 ± 0.9 | 88.5 ± 0.9 | 90.2 ± 0.8 | 80.6 ± 1.1 | 88.9 ± 0.9 | 90.4 ± 0.8 | 87.3 ± 0.9 | 90.0 ± 0.8 | 15.2 ± 1.0 | 91.0 ± 0.8 | 73.9 ± 0.9 | 88.1 ± 0.9 |
| MathQA | 39.0 ± 0.9 | Prompt | 33.5 ± 0.9 | 33.5 ± 0.9 | 32.8 ± 0.9 | 31.5 ± 0.9 | 32.3 ± 0.9 | 33.3 ± 0.9 | 33.6 ± 0.9 | 32.4 ± 0.9 | 33.1 ± 0.9 | 34.1 ± 0.9 | 32.9 ± 0.9 | 33.0 ± 0.9 |
| | | SFT | 43.3 ± 0.9 | 42.6 ± 0.9 | 43.0 ± 0.9 | 43.3 ± 0.9 | 43.2 ± 0.9 | 42.7 ± 0.9 | 42.9 ± 0.9 | 42.9 ± 0.9 | 42.8 ± 0.9 | 43.3 ± 0.9 | 43.0 ± 0.9 | 43.0 ± 0.9 |
| | | DPO | 33.9 ± 0.9 | 34.7 ± 0.9 | 32.9 ± 0.9 | 28.1 ± 0.8 | 30.5 ± 0.8 | 35.0 ± 0.9 | 31.3 ± 0.8 | 32.8 ± 0.9 | 28.9 ± 0.8 | 34.0 ± 0.9 | 31.5 ± 0.8 | 32.9 ± 0.9 |
| *General Reasoning* | | | | | | | | | | | | | | |
| MMLU | 74.5 ± 0.3 | Prompt | 70.3 ± 0.4 | 69.6 ± 0.4 | 40.6 ± 0.4 | 52.8 ± 0.4 | 56.9 ± 0.4 | 72.8 ± 0.4 | 69.0 ± 0.4 | 69.2 ± 0.4 | 55.3 ± 0.4 | 67.9 ± 0.4 | 58.4 ± 0.4 | 66.5 ± 0.4 |
| | | SFT | 72.5 ± 0.4 | 72.0 ± 0.4 | 73.1 ± 0.4 | 68.6 ± 0.4 | 72.1 ± 0.4 | 73.5 ± 0.4 | 72.8 ± 0.4 | 70.7 ± 0.4 | 72.5 ± 0.4 | 73.8 ± 0.4 | 72.6 ± 0.4 | 71.7 ± 0.4 |
| | | DPO | 57.9 ± 0.4 | 64.4 ± 0.4 | 50.3 ± 0.4 | 33.8 ± 0.4 | 42.3 ± 0.4 | 72.3 ± 0.4 | 34.3 ± 0.4 | 62.5 ± 0.4 | 33.2 ± 0.4 | 69.1 ± 0.4 | 43.6 ± 0.4 | 60.4 ± 0.4 |
| GPQA (0-shot) | 33.5 ± 2.2 | Prompt | 31.5 ± 2.2 | 34.2 ± 2.2 | 31.7 ± 2.2 | 32.4 ± 2.2 | 34.6 ± 2.2 | 32.1 ± 2.2 | 32.4 ± 2.2 | 32.8 ± 2.2 | 31.9 ± 2.2 | 32.1 ± 2.2 | 32.4 ± 2.2 | 32.7 ± 2.2 |
| | | SFT | 33.5 ± 2.2 | 32.4 ± 2.2 | 34.2 ± 2.2 | 34.2 ± 2.2 | 33.3 ± 2.2 | 34.4 ± 2.2 | 33.3 ± 2.2 | 33.3 ± 2.2 | 34.4 ± 2.2 | 33.5 ± 2.2 | 33.7 ± 2.2 | 33.6 ± 2.2 |
| | | DPO | 36.8 ± 2.3 | 34.1 ± 2.2 | 35.7 ± 2.3 | 30.6 ± 2.2 | 35.9 ± 2.3 | 35.9 ± 2.3 | 35.5 ± 2.3 | 35.7 ± 2.3 | 32.6 ± 2.2 | 34.6 ± 2.2 | 35.3 ± 2.3 | 33.7 ± 2.2 |
| GPQA (5-shot) | 36.6 ± 2.3 | Prompt | 35.9 ± 2.3 | 32.6 ± 2.2 | 36.2 ± 2.3 | 35.7 ± 2.3 | 36.2 ± 2.3 | 35.7 ± 2.3 | 34.4 ± 2.2 | 34.8 ± 2.3 | 36.6 ± 2.3 | 34.2 ± 2.2 | 35.9 ± 2.3 | 34.6 ± 2.3 |
| | | SFT | 32.4 ± 2.2 | 32.8 ± 2.2 | 34.4 ± 2.2 | 33.7 ± 2.2 | 33.0 ± 2.2 | 33.9 ± 2.2 | 33.7 ± 2.2 | 32.8 ± 2.2 | 33.7 ± 2.2 | 34.8 ± 2.3 | 33.4 ± 2.2 | 33.6 ± 2.2 |
| | | DPO | 37.5 ± 2.3 | 31.2 ± 2.2 | 35.9 ± 2.3 | 31.2 ± 2.2 | 37.1 ± 2.3 | 35.5 ± 2.3 | 33.5 ± 2.2 | 32.1 ± 2.2 | 36.6 ± 2.3 | 35.7 ± 2.3 | 36.1 ± 2.3 | 33.1 ± 2.2 |
| Average | 53.0 ± 1.3 | Prompt | 53.5 ± 1.3 | 52.7 ± 1.3 | 46.3 ± 1.3 | 41.1 ± 1.3 | 50.0 ± 1.3 | 48.3 ± 1.3 | 52.9 ± 1.3 | 50.0 ± 1.3 | 41.0 ± 1.3 | 53.1 ± 1.3 | 48.7 ± 1.3 | 49.1 ± 1.3 |
| | | SFT | 59.1 ± 1.3 | 57.7 ± 1.3 | 59.6 ± 1.3 | 56.2 ± 1.3 | 58.7 ± 1.3 | 59.8 ± 1.3 | 59.2 ± 1.3 | 56.2 ± 1.3 | 57.8 ± 1.3 | 60.0 ± 1.3 | 58.9 ± 1.3 | 58.0 ± 1.3 |
| | | DPO | 53.8 ± 1.3 | 54.7 ± 1.3 | 50.8 ± 1.3 | 41.9 ± 1.3 | 47.0 ± 1.3 | 58.7 ± 1.3 | 47.3 ± 1.3 | 50.7 ± 1.3 | 35.8 ± 1.3 | 55.5 ± 1.3 | 47.0 ± 1.3 | 52.3 ± 1.3 |

Table 11: Benchmark results for different personality traits on `LLaMA-3-70B-Instruct`. **Direct** refers to direct inference without including personality-related prompts. **Prompt** refers to instruction-based prompting. The table includes standard errors (shown as ± values) to provide statistical context for the results.

counterparts in general. This suggests a potential misalignment between DPO's training objectives and the reasoning requirements of specific tasks. These findings highlight the nuanced trade-offs between training strategies, with SFT offering the most reliable approach for balancing personality trait integration and cognitive task performance in large-scale models.

## C.6 REASONING BENCHMARK RESULTS FOR `LLaMA-3-8B-INSTRUCT`

The reasoning evaluation results for the `LLaMA-3-8B-Instruct` model, assessed across six reasoning domains, are summarized in Table 12. Overall, the DPO method generally outperformed SFT and demonstrated performance comparable to the prompt-based approach. This indicates that, with the smaller 8B model, DPO effectively aligns personality traits without significantly compromising reasoning capabilities.

A comparison of personality trait levels revealed that models simulating high trait levels consistently outperformed their low-trait counterparts in both DPO and SFT settings. For instance, on the TruthfulQA benchmark, the high-conscientiousness DPO model achieved 55.0%, significantly surpassing

the low-conscientiousness model's 39.0%. Similarly, on the GSM8K math reasoning task, the high-conscientiousness DPO model scored 72.2%, substantially outperforming the low-level model.

On benchmarks such as TruthfulQA, GPQA (both zero-shot and five-shot), and MathQA, models trained using SFT and DPO performed comparably to the original unaligned model. This suggests that personality trait alignment does not adversely affect reasoning performance in these tasks for a small model. However, notable variations were observed in other benchmarks. For example, DPO exhibited significantly reduced performance on CommonsenseQA and MMLU compared to SFT, prompting, and the original model. Conversely, SFT underperformed on the GSM8K benchmark relative to DPO, prompting, and the original model. These results suggest that the DPO method may be more effective than SFT in preserving or enhancing reasoning performance for specific tasks and traits on small models, though the choice of alignment method may depend on the specific reasoning domain.

| Benchmark | Original | Method | Openness | | Conscientiousness | | Extraversion | | Agreeableness | | Neuroticism | | Average | |
|---|---|---|---|---|---|---|---|---|---|---|---|---|---|---|
| | | | High | Low | High | Low | High | Low | High | Low | High | Low | High | Low |
| *Hallucination Detection* | | | | | | | | | | | | | | |
| **TruthfulQA** | 53.5 | Prompt | 49.0 | 51.5 | 50.6 | 44.4 | 45.3 | 51.9 | 49.2 | 50.3 | 54.6 | 45.2 | 49.7 | 48.7 |
| | | SFT | 50.0 | 45.7 | 50.9 | 43.8 | 46.2 | 52.0 | 49.9 | 46.3 | 53.6 | 42.9 | 50.1 | 46.1 |
| | | DPO | 52.4 | 49.1 | 55.0 | 39.0 | 35.0 | 59.2 | 52.8 | 45.5 | 58.2 | 38.8 | 50.7 | 46.3 |
| *Code Reasoning* | | | | | | | | | | | | | | |
| **HumanEval** | 60.4 | Prompt | 59.1 | 59.8 | 62.2 | 61.6 | 61.0 | 63.4 | 62.8 | 62.2 | 60.4 | 61.6 | 61.1 | 61.7 |
| | | SFT | 57.9 | 54.3 | 59.8 | 56.1 | 58.5 | 57.3 | 60.4 | 54.9 | 58.5 | 58.5 | 59.0 | 56.2 |
| | | DPO | 57.3 | 0.6 | 27.4 | 0.0 | 43.3 | 0.0 | 8.5 | 32.9 | 0.0 | 7.9 | 27.3 | 8.3 |
| **MBPP** | 54.6 | Prompt | 54.6 | 55.4 | 54.2 | 55.2 | 55.8 | 56.0 | 55.4 | 54.8 | 54.4 | 55.8 | 54.9 | 55.4 |
| | | SFT | 56.2 | 56.2 | 54.2 | 56.2 | 56.4 | 56.4 | 55.6 | 55.8 | 55.0 | 56.4 | 55.5 | 56.2 |
| | | DPO | 53.6 | 47.6 | 53.0 | 35.2 | 54.6 | 51.4 | 54.4 | 53.8 | 52.0 | 54.2 | 42.9 | 48.4 |
| *Social Reasoning* | | | | | | | | | | | | | | |
| **SocialIQA** | 49.7 | Prompt | 41.9 | 42.3 | 41.1 | 39.3 | 41.5 | 41.6 | 41.8 | 39.5 | 42.1 | 39.4 | 41.7 | 40.4 |
| | | SFT | 44.0 | 44.9 | 45.9 | 41.9 | 44.4 | 44.6 | 43.7 | 41.4 | 44.6 | 40.8 | 44.5 | 42.7 |
| | | DPO | 43.8 | 43.8 | 42.5 | 37.8 | 41.8 | 40.9 | 42.8 | 38.4 | 42.8 | 39.0 | 42.7 | 40.0 |
| *Commonsense Reasoning* | | | | | | | | | | | | | | |
| **CommonsenseQA** | 51.8 | Prompt | 64.6 | 60.6 | 38.0 | 31.3 | 45.9 | 55.0 | 55.4 | 36.3 | 33.9 | 23.3 | 47.6 | 41.3 |
| | | SFT | 61.8 | 57.9 | 50.5 | 34.3 | 52.7 | 60.8 | 55.4 | 36.0 | 63.4 | 30.6 | 56.8 | 43.9 |
| | | DPO | 22.9 | 24.8 | 48.2 | 21.6 | 29.1 | 56.6 | 28.4 | 26.3 | 47.7 | 23.7 | 35.3 | 30.6 |
| *Math Reasoning* | | | | | | | | | | | | | | |
| **GSM8K** | 64.7 | Prompt | 13.5 | 58.4 | 23.4 | 61.0 | 40.0 | 57.1 | 29.3 | 71.6 | 24.1 | 31.9 | 26.1 | 56.0 |
| | | SFT | 19.8 | 0.5 | 20.2 | 1.4 | 6.0 | 0.5 | 6.4 | 4.8 | 20.1 | 53.3 | 14.5 | 12.1 |
| | | DPO | 68.4 | 31.8 | 72.2 | 31.8 | 69.7 | 63.0 | 70.7 | 64.8 | 71.9 | 3.0 | 70.6 | 38.9 |
| **MathQA** | 27.9 | Prompt | 27.6 | 28.3 | 27.9 | 27.3 | 27.1 | 27.8 | 27.2 | 28.1 | 28.1 | 25.9 | 27.6 | 27.5 |
| | | SFT | 30.1 | 30.2 | 29.6 | 30.3 | 31.0 | 30.6 | 29.6 | 30.3 | 29.6 | 29.4 | 30.0 | 30.2 |
| | | DPO | 26.9 | 27.8 | 28.3 | 25.1 | 25.8 | 27.6 | 24.9 | 27.7 | 29.7 | 24.9 | 27.1 | 26.6 |
| *General Knowledge* | | | | | | | | | | | | | | |
| **MMLU** | 51.2 | Prompt | 37.5 | 29.1 | 23.2 | 27.0 | 24.7 | 29.2 | 27.7 | 25.5 | 23.4 | 23.8 | 27.3 | 26.9 |
| | | SFT | 45.0 | 48.5 | 35.6 | 32.0 | 37.5 | 46.5 | 44.2 | 39.9 | 47.1 | 31.7 | 41.9 | 39.7 |
| | | DPO | 23.0 | 29.8 | 29.7 | 26.9 | 24.8 | 41.4 | 30.7 | 26.3 | 30.8 | 23.1 | 27.8 | 29.5 |
| **GPQA (0-shot)** | 28.1 | Prompt | 29.0 | 28.8 | 28.6 | 23.0 | 28.6 | 29.2 | 29.0 | 27.2 | 28.8 | 28.3 | 28.8 | 27.3 |
| | | SFT | 27.9 | 27.9 | 28.1 | 25.0 | 27.2 | 28.3 | 28.8 | 24.1 | 29.0 | 28.3 | 28.2 | 26.7 |
| | | DPO | 27.9 | 25.0 | 29.7 | 21.0 | 27.2 | 26.8 | 28.8 | 21.4 | 29.5 | 25.2 | 28.6 | 23.9 |
| **GPQA (5-shot)** | 29.9 | Prompt | 29.7 | 26.6 | 28.8 | 26.8 | 28.3 | 26.6 | 27.9 | 28.6 | 29.0 | 25.2 | 28.7 | 26.8 |
| | | SFT | 26.1 | 27.0 | 28.8 | 26.6 | 28.8 | 28.6 | 30.6 | 27.9 | 28.6 | 27.5 | 28.6 | 27.5 |
| | | DPO | 27.9 | 26.3 | 28.3 | 23.0 | 26.8 | 28.1 | 27.5 | 24.6 | 28.8 | 25.2 | 27.9 | 25.4 |
| **Average** | 43.9 | Prompt | 35.8 | 40.5 | 31.5 | 34.4 | 34.3 | 39.5 | 35.1 | 38.2 | 31.7 | 29.1 | 33.7 | 36.4 |
| | | SFT | 37.2 | 34.0 | 34.8 | 27.6 | 32.8 | 35.3 | 35.0 | 29.9 | 38.8 | 34.8 | 35.7 | 32.3 |
| | | DPO | 35.6 | 30.7 | 41.6 | 26.9 | 34.1 | 43.2 | 37.7 | 33.8 | 42.4 | 23.4 | 38.3 | 31.6 |

Table 12: Benchmark results for the `LLaMA-3-8B-Instruct` model are presented across various personality traits and evaluation methods. The benchmarks are categorized into six key areas: Hallucination Detection, General Reasoning, Social Reasoning, Commonsense Reasoning, Mathematical Reasoning, and General Knowledge.

## D  CORRELATION BETWEEN PERSONALITY TRAITS AND REASONING BEHAVIORS

### D.1  HUMAN VS. LLaMA-3-70B-Instruct

Understanding the influence of personality traits on reasoning behaviors in LLMs is crucial for developing models tailored to specific personality profiles. Research on the Big Five personality traits has consistently demonstrated their significant impact on human cognition and problem-solving abilities (John et al., 1999; Soto et al., 2011). Traits such as openness, conscientiousness, and agreeableness are often associated with enhanced reasoning capabilities, while neuroticism has been found to impair performance across a range of reasoning tasks (Ackerman & Heggestad, 1997; Schaie et al., 2004; Chamorro-Premuzic et al., 2006).

Table 13 summarizes relevant findings from recent psychological studies and their alignment with our experimental results on LLaMA-3-70B-Instruct. Our findings corroborate these studies, indicating that models exhibiting higher conscientiousness and agreeableness generally perform better in reasoning tasks. In contrast, models characterized by lower levels of extraversion and neuroticism also demonstrate improved reasoning performance. These results highlight the potential of personality-aligned training to optimize LLM performance for reasoning-intensive tasks.

---

**Openness** Openness is associated with intellectual curiosity and creativity and enhances problem-solving in tasks requiring abstract reasoning and social cognition (Ackerman & Heggestad, 1997; McCrae, 1987). While research indicates that openness positively correlates with cognitive abilities (Chamorro-Premuzic et al., 2006; Costa Jr et al., 1976; Graham & Lachman, 2012; Schaie et al., 2004), our models do not show significant performance differences across reasoning tasks based on openness levels, with the exception of SFT on math reasoning tasks. This suggests that openness may not directly translate to gains in reasoning tasks beyond math, despite its known benefits to human cognition.

---

**Conscientiousness** Conscientiousness, linked to discipline and organization, consistently improves model performance in mathematical reasoning and hallucination detection. This aligns with psychological studies showing that higher conscientiousness is linked to better academic performance and fewer errors in cognitive tasks due to increased diligence and thoroughness (Roberts et al., 2014; Poropat, 2009; Digman, 1990; Moutafi et al., 2003; Schaie et al., 2004).

---

**Extraversion** Extraversion is often associated with sociability and shows mixed results in cognitive tasks. While it can enhance social reasoning, it may negatively affect individual problem-solving tasks, such as math reasoning (Blickle, 1996; Ashton et al., 2002; Costa Jr et al., 1976). Our models simulating lower extraversion perform better across many reasoning domains, including math and also commonsense reasoning, consistent with findings that high extraversion can detract from tasks requiring focused, solitary work (Matthews & Gilliland, 1999; Chamorro-Premuzic & Furnham, 2006).

---

**Agreeableness** Agreeableness, linked to traits like trust and cooperation, improves social reasoning in our models, consistent with human studies (Graziano, 1997). However, it shows minimal impact on math or commonsense reasoning, reflecting research suggesting that agreeableness is less beneficial for analytical tasks (Poropat, 2009; Ackerman & Heggestad, 1997; Schaie et al., 2004).

---

**Neuroticism** Neuroticism reflects emotional instability, and is consistently associated with poorer cognitive performance due to anxiety and cognitive interference, especially social reasoning and hallucination detection (Robinson & Tamir, 2005; Zeidner, 2005; Chamorro-Premuzic et al., 2006; Eysenck, 2013). Our models confirm this, with lower Neuroticism levels leading to better performance across almost all reasoning tasks.

---

Table 13: Summary of the influence of Big Five personality traits on reasoning tasks in human cognition, and comparison of psychological research findings with our experimental results on LLMs.

## D.2 Human vs. LLaMA-3-8B-Instruct

The influence of Big Five Personality traits on reasoning tasks in human cognition, as outlined in Table 13, served as a foundation for analyzing the performance of the LLaMA-3-8B-Instruct model. This analysis aims to explore how alignment with different personality traits affects the model's reasoning capabilities. Below, we summarize the observed correlations between each trait and the model's performance across various reasoning benchmarks.

**Openness** The impact of Openness on reasoning performance was highly task-dependent. Models aligned with high levels of Openness using the DPO method exhibited significantly improved performance in mathematical reasoning tasks. However, these models underperformed in commonsense reasoning benchmarks compared to both the prompt-based approach and the original model. These results suggest that while high Openness alignment enhances mathematical reasoning, it does not guarantee consistent improvements across all reasoning domains.

**Conscientiousness** A strong positive correlation was observed between Conscientiousness and reasoning performance. Models aligned with higher levels of Conscientiousness consistently outperformed their low-level counterparts across most benchmarks. This trend highlights that high Conscientiousness alignment likely enhances systematic reasoning and attention to detail, benefiting performance across diverse reasoning tasks.

**Extraversion** Lower levels of Extraversion were associated with better performance across reasoning tasks. Specifically, in commonsense reasoning benchmarks, models with low Extraversion significantly outperformed those with high Extraversion. This negative correlation suggests that high Extraversion may introduce distractibility, potentially impeding performance in tasks that require focused attention and analytical reasoning.

**Agreeableness** The influence of Agreeableness on reasoning performance was minimal and inconsistent. No clear advantage was observed for models aligned with either high or low levels of Agreeableness across the benchmarks. These findings indicate that Agreeableness has a weak correlation with the model's reasoning capabilities, suggesting its alignment has little effect on overall performance.

**Neuroticism** The relationship between Neuroticism and reasoning performance was inconsistent and did not align with expectations from human cognition studies. High Neuroticism models performed well in some reasoning tasks, while low Neuroticism models scored poorly in others. These results imply that high Neuroticism alignment does not necessarily impair reasoning performance, contrasting with psychological findings in humans. This discrepancy may arise from limitations in how Neuroticism is modeled and represented in the training process.

