# OpenReview forum: "BIG5-CHAT: Shaping LLM Personalities Through Training on Human-Grounded Data"
_ICLR.cc/2025/Conference — ICLR 2025 Conference Withdrawn Submission_

### Official Review · Reviewer_nFT1 · 2024-11-03

**Soundness:** 3
**Presentation:** 4
**Contribution:** 2
**Rating:** 5
**Confidence:** 5

**Summary:**

The paper presents BIG5-CHAT, a dataset of 100,000 dialogues for simulating realistic human personality traits. The authors leverage two datasets, PsychGenerator and SODA, to generate BIG5-CHAT with DExperts framework and train LLMs on realistic personality patterns. The experiments show that training-based methods outperform prompting methods on personality assessments. Other experiments also reveal the personality traits of LLMs have influence on behaviors in various reasoning tasks, which align with human cognitive tendencies observed in psychology.

**Strengths:**

Originality: The BIG5-CHAT dataset is novel and represents a valuable resource for advancing human-like LLM research.

Quality: The experiments are well-designed and comprehensive. The results on several benchmarks are also aligned well with psychology researches.

Clarity: The paper is easy to follow, with clear explanations of the methods and concepts.

Significance: The paper provides a new dialogue dataset with personality traits. Some findings show the impacts of personality traits in different benchmarks.

**Weaknesses:**

Absence of Human Evaluation: To enhance the validity of personality alignment claims, human evaluations are recommended. A suggested methodology is to have a panel of human raters assess a sample of dialogues from BIG5-CHAT, scoring them on each of the Big Five traits (Openness, Conscientiousness, Extraversion, Agreeableness, Neuroticism). The human ratings can then be compared to the intended trait levels to verify if the model successfully conveys the intended personality traits.

Limited Novelty: Although this paper introduces the BIG5-CHAT dataset and applies training methods like SFT and DPO for personality alignment, prior work has shown that it is possible to train LLMs to exhibit specific personality traits, e.g., https://arxiv.org/pdf/2312.12999. In this paper mentioned above, they collected data of different MBTI types and trained 16 agents. These agents can chat in the way of certain personality trait and have the ability to pass the MBTI test. Additionally, the method of generating personality-aligned responses is adapted from DExperts. To strengthen the novelty, it would be helpful if the authors provided a clearer comparison to existing work, particularly in how their dataset creation or methodology differs. Emphasizing unique aspects, such as the integration of PsychGenerator and SODA datasets or specific adjustments to the DExperts framework for personality traits, would clarify the innovative contributions of this study.

**Questions:**

Question 1: How do you interpret the results of Table 1, in which the proposed method is not as effective as two baseline models in "Agreeableness" and "Neuroticism"? What makes the difference between these traits?

Question 2: How do you determine the scaling factor \gamma? How much does this factor influence the generated results?

---

> ### Author Response · Authors · 2024-11-23
> **Response to Reviewer nFT1 (1/n)**
>
> Thank you for the questions. Please consider raising the score if we have addressed your concerns well.
> >Q1: Absence of human evaluation
>
> We appreciate the reviewer’s suggestion to include human evaluation. We have conducted two human evaluation studies as follows:
>
> (1) Human Evaluation for the Big5-Chat Dataset. We agree that human evaluation is a crucial method to assess the realism and validity of Big5-Chat. To address this, we conducted a human evaluation to compare our dataset to a Llama-3-70B-Instruct baseline, which uses the same procedure for generating dialog responses but does not incorporate expert generators or the DExperts framework. In this baseline, the personality traits are induced through the prompt: “You are a person with the following Big Five personality trait: {trait} - {level}.” Please refer to Appendix C.1 for the details about this human evaluation setup. **TLDR: Our model outperforms the Llama-3-70B-Instruct baseline in this human evaluation in both the expressiveness of the expected personality trait level and the realism of social interactions.** Since other LLM personality datasets are created by prompting LLMs to generate synthetic personality data, this suggests that other datasets are less realistic than Big5-Chat. The human evaluation results are as follows:
>
> 1. Comparison with the Llama-3-70B-Instruct baseline in expressiveness of personality trait level: 50.3% of the time, the expert generator was rated better; 39.8% of the time, the generator and the baseline are rated equal; 10.0% of the time, expert model was rated worse than the baseline. The annotator agreement is 0.50 using Cohen’s Kappa coefficient.
> 2. Comparison with the Llama-3-70B-Instruct baseline in realism: 47.8% of the time, the expert generator was rated better; 42.3% of the time, both the generator and the baseline are rated equal; 10.0% of the time, expert model was rated worse than the baseline. The annotator agreement is 0.55 using Cohen’s Kappa coefficient.
>
>   These results demonstrate that DExpert significantly outperformed the prompting baseline in terms of both realism and the expressiveness of personality levels, as validated by human judgment. This emphasizes the limitations of prompt-based approaches, which rely on general-purpose models and often lack the fine-grained, human-grounded control necessary for nuanced personality expression.
>
> In fact, we critically examine existing personality infusion methods, specifically the common practice of prompting with behavior descriptions. Our argument is that such prompting approaches lack grounding in how personality is naturally expressed through language. Instead, they rely on predefined behavioral descriptions that LLMs inherently do not possess or authentically embody. Please refer to Line 357 about the demonstration prompt setup to understand the limitations of this line of approach.
>
> (2) Human Evaluation for the Expert Generator. We also had a deeper analysis of the human annotation scores across personality traits for the topic-post generation baseline. We introduced an additional baseline designed to mirror the expert generator's post-generation strategy. This new baseline prompts gpt-4o-mini to complete a Facebook post given the first five words and the target personality traits, while also incorporating the post format for guidance. The prompt used for this baseline is a slightly modified version of the expert generator training instructions described in Section 3.2 and has been added to Appendix B.1. Below, we first describe the human evaluation setup and results, and then summarize the relevant findings. This human evaluation is set up to compare the expert generator with the two baselines mentioned in Table 1 for a more human-grounded assessment. Two graduate students familiar with the Big Five personality framework were asked to evaluate two separate sets of comparisons:
>
>  1. Expert generator outputs vs. posts generated using the topic-post generation baseline.
>
>  2. Expert generator outputs vs. posts generated using the post-completion baseline.
>
> Please refer to the newly added Appendix C.2 for details on the human evaluation setup. The human evaluation results are as follows:
>
> 1. Comparison with the original baseline: 66.5% of the time, the expert generator was rated better than the original baseline. The inter-annotator agreement is 0.61 using Cohen’s Kappa coefficient. Qualitatively, this baseline fails to generate realistic text, as it overuses emojis and includes exaggerated expressions.
>
> 2. Comparison with the post-completion baseline: 79.25% of the time, the expert generator was rated better than the post-completion baseline. The inter-annotator agreement is 0.41 using Cohen’s Kappa coefficient.

---

> ### Author Response · Authors · 2024-11-23
> **Response to Reviewer nFT1 (2/n)**
>
> Continued in Q1:
> >Q1: Absence of human evaluation
>
> These results show that the expert generator was judged to be more capable of expressing personality traits than baselines. Furthermore, the lower classifier accuracy and human evaluation ratings for the post-completion baseline suggest that the challenges imposed by the expert generator’s strategy make it harder to match the desired traits, underscoring the validity of the classifier’s assessment. While these results should be interpreted cautiously as our human evaluators are not psychological experts, this can still provide support to our argument that the expert generator is more capable of expressing personality traits in a grounded, realistic way. Since the post-generation baseline is newly added, we have incorporated descriptions of the post-generation baseline and its results in Section 4.3 and Appendix B.1.
>
> Therefore, based on our thorough and rigorously designed human evaluation, combined with the updated results presented in Table 6, we believe the evidence strongly supports the validity of the dataset and our claims. We welcome further discussion and are happy to address any additional questions the reviewer may have.
>
> > Q2: Limited novelty
>
> Thank you for the question. While prior works have claimed the ability of LLMs to mimic personality traits, our approach offers several unique contributions that distinguish it from existing work:
>
> (1) **Human-grounded dataset and methodology.** Unlike previous work such as Machine Mindset [1], which generates personality datasets solely by prompting LLMs, our method is grounded in human-authored texts. Specifically, we utilize a human-grounded expert generator during the decoding process to imbue models with specific personality traits. Compared to LLM prompting, our approach induces personality traits more effectively, expressively, and realistically, as evidenced by both psychometric tests and human evaluations. Notably, Big5-Chat is the first large-scale dataset specifically designed to capture realistic expressions of the Big Five personality traits in open-ended dialogues. To our knowledge, no comparable datasets exist in this space. The newly added Table 6 provides a detailed comparison of Big5-Chat with other datasets.
>
> (2) **Novel approach to inducing personality in dialogue.** Our method induces personality traits by leveraging human-written data (via PsychGenerator) and diverse open-ended dialogue scenarios (via SODA). Unlike prior work, we transfer personality expressions from human-written data to create more grounded social interaction data. This methodology allows us to generate diverse and contextually relevant personality-aligned examples from existing human-written sources.
>
> (3) **Enhanced generalizability in social interactions.** By combining personality-specific experts trained on PsychGenerator with the socially rich SODA dataset, our approach achieves greater generalizability across social scenarios. SODA encompasses a wide range of emotions and thematic keywords, enabling our model to handle complex conversational dynamics. As demonstrated in the original DExperts paper, the DExperts framework mitigates domain constraints by interpolating between the base LLM and domain-specific experts. This property is particularly beneficial in leveraging PsychGenerator's human-grounded data. Section 4.2 elaborates on Big5-Chat's dataset composition and its implications.
>
> (4) **Exploring personality’s impact on reasoning.** To our knowledge, we are the first to explore how personality traits in LLMs affect reasoning behaviors and performance. Our study uncovers alignments between personality-driven models and human psychological research, providing a foundation for future investigations into how traits influence cognitive abilities. These findings underscore the broader potential of personality-aligned LLMs in reasoning tasks. For further details, refer to Section 2 (paragraph starting with “Unintended influence on reasoning patterns”), Section 5.3, and Appendices C and D.
>
> [1] Cui, Jiaxi, et al. "Machine mindset: An mbti exploration of large language models." arXiv preprint arXiv:2312.12999 (2023).

---

> ### Author Response · Authors · 2024-11-23
> **Response to Reviewer nFT1 (3/n)**
>
> Continued in Q2:
> > Q2: Limited novelty
>
> (5) **Advancing beyond MBTI-based models.** While prior works like the referenced MBTI-based study train models on personality-aligned dialogues, it is important to note that the MBTI framework has been widely criticized as pseudoscientific [2][3]. In contrast, the Big Five model is more scientifically robust and widely validated [4][5]. Section 2 outlines the advantages of the Big Five over other personality models.
>
> [2] Stein, Randy, and Alexander B. Swan. "Evaluating the validity of Myers‐Briggs Type Indicator theory: A teaching tool and window into intuitive psychology." Social and Personality Psychology Compass 13.2 (2019): e12434.
>
> [3] Schweiger, David M. "Measuring managerial cognitive styles: On the logical validity of the Myers-Briggs Type Indicator." Journal of Business Research 13.4 (1985): 315-328.
>
> [4] Gardner, William L., and Mark J. Martinko. "Using the Myers-Briggs Type Indicator to study managers: A literature review and research agenda." Journal of management 22.1 (1996): 45-83.
>
> [5] Pittenger, David J. "Cautionary comments regarding the Myers-Briggs type indicator." Consulting Psychology Journal: Practice and Research 57.3 (2005): 210.
>
> > Q3: Interpretation of Table 1 results
>
> We thank the reviewer for pointing this question out. The question prompted us to have a deeper analysis of the human annotation scores across personality traits, as detailed in the “Human Evaluation for the Expert Generator” section in Answer 1 for the topic-post generation baseline. Below, we summarize the relevant findings:
>
> For annotator 1, the ratios where the expert generator outperformed GPT were 77.5% for openness, 70.0% for conscientiousness, 75.0% for extraversion, 67.5% for agreeableness, and 67.5% for neuroticism. For annotator 2, the ratios where the expert generator outperformed GPT were 87.5% for openness, 90.0% for conscientiousness, 95.0% for extraversion, 80.0% for agreeableness, and 82.5% for neuroticism.
>
> These human annotation results notably align with the classifier results in Table 1, where agreeableness and neuroticism exhibit the worst results. This suggests that these traits present challenges for both classifier-based and human-grounded personality assessments. We hypothesize that the observed differences across traits stem from intrinsic properties of agreeableness and neuroticism. Specifically, the agreeableness trait may require nuanced interpersonal context and subtle emotional cues, which can be harder for both models and annotators to consistently capture and evaluate; the neuroticism trait often involves complex expressions of emotional instability or stress, which may vary widely across instances and pose difficulties in generating or interpreting consistent patterns. Although the topic-post generation baseline is able to "fool" the classifier, the human evaluation highlights its limitations in aligning with human-grounded personality induction.
>
> > Q4: How did you select the value for the scaling factor \gamma
>
> We determined the scaling factor $\gamma$ through a qualitative evaluation conducted on a curated evaluation set of 50 examples. We tested a range of values for $\gamma$ from \{0.25, 0.5, 1.0, 2.0, 4.0\} and identified the following patterns:
>
> -	When \($\gamma$ > 0.5\), the generated responses often fail to align with the posed questions or conform to the intended social scenarios. These outputs also exhibited issues such as repetitive phrasing and a noticeable decline in coherence when following instructions. This indicates that assigning too much weight to the generator undermines the model’s ability to adhere to task instructions effectively.
>
> -	When \($\gamma$ < 0.5\), the responses resembled generic outputs from large language models, characterized by unnatural phrasing and verbosity. These outputs often included redundant sentences or excessive formalities, such as unnecessary greetings, leading to less concise, less engaging, and less diverse dialogues.
>
> Based on this analysis, we selected \($\gamma$ = 0.5\) as the optimal value to balance the instruction-following ability, linguistic naturalness, and conversational diversity. Although detailed ablation studies were not included in the paper, this analysis demonstrates the clear and significant impact of $\gamma$ on the quality of generated results.

---

> ### Author Response · Authors · 2024-11-25
>
> Dear Reviewer nFT1,
>
> As the discussion period comes to a close, we wanted to follow up to ensure our response has fully addressed your concerns. If there are any additional points you’d like us to clarify or elaborate on, we’d be happy to provide further details. We truly appreciate the time and effort you’ve dedicated to reviewing our work, and we look forward to hearing from you soon.

---

### Official Review · Reviewer_xgRk · 2024-11-03

**Soundness:** 3
**Presentation:** 3
**Contribution:** 2
**Rating:** 5
**Confidence:** 5

**Summary:**

Challenges:
1. Existing methods rely on prompting models with descriptions of behaviors associated with personality traits. These behavioral descriptions are nonsensical for text-based LLMs, failing to ground their personality in realistic patterns of how humans’ personality is expressed in text.
2. The scarcity of large-scale, human-generated datasets annotated with personality traits has hindered the exploration of training-based approaches, limiting most prior research to prompting-based methods.

To address these challenges, this paper introduces BIG5-CHAT, a large-scale dialogue responses dataset that is designed to capture Big Five personality traits within diverse social interactions. Also, leveraging the dataset, this work empirically investigates if training-based methods benefit in grounding in real human data compared to traditional prompting methods.

**Strengths:**

- The paper is clearly written and easy to understand.
- Background section is thorough and well-written that helps the reviewers to understand more deeply about the domain and the current challenges.

**Weaknesses:**

- **Although the motivation for the work is convincing, contribution is comparatively trivial.** Much of their dataset and framework are adopted from previous works, with only slight change. Also, the necessity of conducting alignment comparison experiment only on their dataset is unclear. Is there a specific reason or intention as to why the authors did not experiment on existing datasets?
- **The argument that BIG5-CHAT captures personality traits better than other datasets is not convincing.** In line 265, the authors argue their dataset captures personality and nuances better than other datasets. However, the provided table (Table 8) does not demonstrate the argument. Human evaluation, or at the very least, case study for comparing BIG5-CHAT over other datasets is required. Also in experiments, the authors did not conduct experiments on other datasets, leaving the validity of the dataset in question.
- **This work fails to capture all traits comprehensively.** The proposed method, DExperts, induces each personality trait in a response. However, in reality, the personality of a human is very complex and comprehensive. It is doubtful that their approach, as opposed to their statement of “capturing nuanced expression of personality traits”, can truly reflect human personality more elaborately.
- **Related work section could include more works.** Although a considerable portion of the work deals with alignment methods, review on works that concerns effectiveness of training-based methods in inducing personality, or let alone works on alignment, are nowhere to be seen.

**Questions:**

- line 363-367: Does this mean that personality traits are evaluated by humans using questionnaires? If so, the authors should provide details on evaluator recruitment.
- (Related to third weakness) Have the authors considered combining all traits in a single response, like combining all five expert logits into one combined logit (in Equation 1)?
- line 420: The authors said that they excluded results for instruction-based prompting on Llama-3-8B-Instruct. However, Table 2 reports Prompt-Inst and excludes Prompt-Demo. Which is correct? There must be a clarification on this.
- Experiments: In my opinion, experimenting SFT, DPO and prompting on other datasets (the ones that you mentioned as failing to ground human personality realistically) is necessary to prove the effectiveness of training-based methods for inducing personality traits. In that case, the authors will be able to demonstrate the effectiveness of training methods, as well as excellence of their dataset over others.

---

> ### Author Response · Authors · 2024-11-23
> **Response to Reviewer xgRk (1/n)**
>
> Thank you for the questions. Please consider raising the score if we have addressed your concerns well.
>
> > Q1: Contribution is trivial; Necessity of using Big5-Chat
>
> We are grateful for the reviewer's recognition of the motivation behind our work. Regarding the concerns about the perceived trivial nature of our contribution and the reliance on an adapted dataset and framework, we would like to emphasize the novelty and necessity of Big5-Chat:
> 1. **First human-grounded dataset in the Big Five personality domain**: Big5-Chat is the first large-scale dataset specifically designed to capture realistic expressions of the Big Five personality traits. Unlike prior datasets, it incorporates real human language use grounded in expert personality annotation, addressing the lack of genuine, human-grounded data for training LLMs to express different personalities in dialog data. To the best of our knowledge, there are no other such personality datasets available. Existing datasets (as shown in Table 6) either rely on purely synthetic data or proxy metrics that fail to capture authentic personality expressions. Big5-Chat addresses this gap by grounding the generated dialogues in real personality annotations, a critical feature for realistic personality simulation.
> 2. **Necessity of our dataset: While the reviewer suggests experimenting on existing datasets, these datasets do not align with the Big Five framework or lack the features needed to evaluate our approach comprehensively.** As shown in Table 6, recent works focus predominantly on questionnaires, which fail to provide natural conversational language that reflects personality traits in an authentic and grounded manner. As discussed in Section 2, datasets derived from psychometric test questionnaires lack validity for this purpose. In contrast, Big5-Chat offers realistic interaction scenarios, capturing how personality traits are genuinely expressed in human communication, without relying on pre-existing questionnaires. Datasets like HP dataset (in Table 6) are designed for character estimate domains but lack nuanced personality annotations.
>
>
> > Q2: Lack of human evaluation and experiments on other datasets
>
> As discussed in response to Q1, Big5-Chat is the first large-scale dataset specifically designed to capture realistic expressions of the Big Five personality traits in open-ended dialogues. To the best of our knowledge, no comparable datasets currently exist that focus on personality traits in this manner. We have updated Table 6 to highlight the differences between Big5-Chat and previously available datasets, providing a clearer context for its unique contributions.
>
> However, we agree that human evaluation is a crucial method to assess the realism and validity of Big5-Chat. To address this, we conducted a human evaluation to compare our dataset to a Llama-3-70B-Instruct baseline, which uses the same procedure for generating dialog responses but does not incorporate expert generators or the DExperts framework. In this baseline, the personality traits are induced through the prompt: “You are a person with the following Big Five personality trait: {trait} - {level}.” Please refer to Appendix C.1 for the details about this human evaluation setup. **TLDR: Our model outperforms the Llama-3-70B-Instruct baseline in this human evaluation in both the expressiveness of the expected personality trait level and the realism of social interactions.** Since other LLM personality datasets are created by prompting LLMs to generate synthetic personality data, this suggests that other datasets are less realistic than Big5-Chat. The human evaluation results are as follows:
> 1. Comparison with the Llama-3-70B-Instruct baseline in expressiveness of personality trait level: 50.3% of the time, the expert generator was rated better; 39.8% of the time, the generator and the baseline are rated equal; 10.0% of the time, expert model was rated worse than the baseline. The annotator agreement is 0.50 using Cohen’s Kappa coefficient.
> 2. Comparison with the Llama-3-70B-Instruct baseline in realism: 47.8% of the time, the expert generator was rated better; 42.3% of the time, both the generator and the baseline are rated equal; 10.0% of the time, expert model was rated worse than the baseline. The annotator agreement is 0.55 using Cohen’s Kappa coefficient.
>
> These results demonstrate that our approach significantly outperformed the prompting baseline in terms of both realism and the expressiveness of personality levels, as validated by human judgment. This emphasizes the limitations of prompt-based approaches, which rely on general-purpose models and often lack the fine-grained, human-grounded control necessary for nuanced personality expression.

---

> ### Author Response · Authors · 2024-11-23
> **Response to Reviewer xgRk (2/n)**
>
> Continued in Q2:
>
> > Q2: Lack of human evaluation and experiments on other datasets
>
> In fact, we critically examine existing personality infusion methods, specifically the common practice of prompting with behavior descriptions. Our argument is that such prompting approaches lack grounding in how personality is naturally expressed through language. Instead, they rely on predefined behavioral descriptions that LLMs inherently do not possess or authentically embody. Please refer to Line 357 about the demonstration prompt setup to understand the limitations of this line of approach.
>
> Therefore, based on our thorough and rigorously designed human evaluation, combined with the updated results presented in Table 6, we believe the evidence strongly supports the validity of the dataset and our claims. We welcome further discussion and are happy to address any additional questions the reviewer may have.
>
> > Q3:  Single-trait focus fails to reflect human personality comprehensively. Did the authors attempt to mix traits?
>
> Our method is not inherently limited to steering only one trait at a time. In fact, the proposed Dexpert framework is naturally extensible to multi-trait steering. By combining logits from multiple expert models—each corresponding to a distinct personality trait—during decoding, our approach can generate text that reflects blended personality profiles. Furthermore, methods like those proposed in Machine Mindset [1], which concatenate training data across different traits, can complement this capability to induce multiple traits simultaneously.
>
> In this study, however, we deliberately focus on isolating individual traits. This choice was motivated by a desire to clearly understand the behaviors associated with specific traits in isolation. Isolating traits enhances the clarity, interpretability, and replicability of our findings, which aligns with established practices in personality modeling research [2].
>
> We agree with the reviewer’s observation that traits are often correlated and rarely appear in isolation, while we have included an analysis in Appendix C.2 and Figure 2 that explores the correlations between different traits in both model-generated and human outputs, providing initial insights into the interconnectedness of personality traits.
>
> We also recognize the importance of exploring multi-trait interactions and have added a discussion in the paper on how our methodology could be extended by blending multiple expert models. This extension is indeed a promising direction for future work and we have added it to Section 8. In subsequent studies, we plan to explore the simultaneous induction of multiple traits to investigate more nuanced personality dynamics in text generation. However, this is beyond the scope of this work.
>
> [1] Cui et al., Machine Mindset: An MBTI Exploration of Large Language Models
>
> [2] Jiang et al., Evaluating and Inducing Personality in Pre-trained Language Models
>
> > Q4: The related work section is incomplete
>
> We appreciate the reviewer’s feedback regarding the scope of the related work section. In Section 6 of our paper, we have made an effort to include recent and relevant works specifically addressing the effectiveness of training-based methods for inducing personality traits in language models, particularly focusing on the Big Five personality framework.
> While our research does not cover general-purpose alignment methods, as they fall outside the scope of this work, we acknowledge that alignment is a broad and important field. Given the practical limitations of including all related studies, we aimed to prioritize works most directly relevant to our research domain.
>
> **If there are specific works on Big Five personality induction or closely related areas that you believe should be included but are currently omitted, we kindly ask you to share those references, and we will be happy to address them in the revised version of our paper.**
>
> > Q5: Are personality traits evaluated by humans using questionnaires?
>
> We do not employ human evaluators to assess personality traits through questionnaires. Instead, we use established questionnaires from the BFI [3] and IPIP-NEO [4] frameworks as prompts to automatically evaluate the personality traits induced in language models.
>
> We believe there might have been a misunderstanding regarding the term "crowd-sourced" in the original draft. This term refers to the way of deriving the crowd scores presented in Figure 2, not to the involvement of human evaluators in assessing traits. To address this ambiguity, we have revised the wording in Section 5.1 and Appendix C.3 to enhance clarity.
>
> [3] Huang et al., Who is ChatGPT? Benchmarking LLMs' Psychological Portrayal Using PsychoBench
>
> [4] Jiang et al., Evaluating and Inducing Personality in Pre-trained Language Models

---

> ### Author Response · Authors · 2024-11-23
> **Response to Reviewer xgRk (3/n)**
>
> > Q6: Clarifications on Instruction-Based prompting in Table 2
>
> Thank you for pointing this out and we apologize for the typo. To clarify, we excluded demonstration-based prompting due to limited instruction-following capabilities in the LLaMA-3-8B-Instruct model. We have fixed it in the PDF submission.
>
> > Q7: Compare your approach with other baselines
>
> In addition to the discussion in Q2, we have also conducted another human evaluation to compare the expert generator with the two baselines mentioned in Table 1 for a more human-grounded assessment. We introduced an additional baseline designed to mirror the expert generator's post-generation strategy. This new baseline prompts gpt-4o-mini to complete a Facebook post given the first five words and the target personality traits, while also incorporating the post format for guidance. The prompt used for this baseline is a slightly modified version of the expert generator training instructions described in Section 3.2 and has been added to Appendix B.1. Two graduate students familiar with the Big Five personality framework were asked to evaluate two separate sets of comparisons:
> 1. Expert generator outputs vs. posts generated using the topic-post generation baseline.
> 2. Expert generator outputs vs. posts generated using the post-completion baseline.
>
> Please refer to the newly added Appendix C.2 for details on the human evaluation setup. The human evaluation results are as follows:
> 1. Comparison with the original baseline: 66.5% of the time, the expert generator was rated better than the original baseline. The inter-annotator agreement is 0.61 using Cohen’s Kappa coefficient. Qualitatively, this baseline fails to generate realistic text, as it overuses emojis and includes exaggerated expressions.
> 2. Comparison with the post-completion baseline: 79.25% of the time, the expert generator was rated better than the post-completion baseline. The inter-annotator agreement is 0.41 using Cohen’s Kappa coefficient.
>
> These results show that the expert generator was judged to be more capable of expressing personality traits than baselines. Furthermore, the lower classifier accuracy and human evaluation ratings for the post-completion baseline suggest that the challenges imposed by the expert generator’s strategy make it harder to match the desired traits, underscoring the validity of the classifier’s assessment. While these results should be interpreted cautiously as our human evaluators are not psychological experts, this can still provide support to our argument that the expert generator is more capable of expressing personality traits in a grounded, realistic way. Since the post-generation baseline is newly added, we have incorporated descriptions of the post-generation baseline and its results in Section 4.3 and Appendix B.1.

---

> ### Author Response · Authors · 2024-11-25
>
> Dear Reviewer xgRk,
>
> As the discussion period comes to a close, we wanted to follow up to ensure our response has fully addressed your concerns. If there are any additional points you’d like us to clarify or elaborate on, we’d be happy to provide further details. We truly appreciate the time and effort you’ve dedicated to reviewing our work, and we look forward to hearing from you soon.

---

> > ### Comment · Reviewer_xgRk · 2024-11-26
> >
> > Thank you to the authors for the thoughtful revisions and detailed responses. After carefully reviewing your updates and explanations, I have decided to maintain my original score.

---

> ### Author Response · Authors · 2024-11-26
>
> Dear Reviewer xgRk,
>
> Thank you for reviewing our revisions. We would greatly appreciate it if you could elaborate on the specific aspects of our work that remain a concern. If there are additional steps we could take to address these issues or improve our work, we would be eager to hear your suggestions. Your insights are invaluable in helping us better understand your perspective and enhance our work.

---

### Official Review · Reviewer_xx4V · 2024-11-03

**Soundness:** 3
**Presentation:** 2
**Contribution:** 3
**Rating:** 6
**Confidence:** 3

**Summary:**

The paper tackles the problem of embedding personality traits into LLMs. To this end, the authors propose:
- A novel large-scale dataset, BIG5-CHAT, of 100000 dialogues which grounds real human personalities in text. To generate the dataset they combine controllable text generation models with a domain-specific, personality-annotated dataset. They propose a classification-based evaluation to address to which extent the proposed generation framework is capable of embedding the personality traits within the text.
- The authors test the BIG5-CHAT over two instruction-based models, i.e. Llama-3-8B and Llama-3-70B under three main learning strategies: traditional prompting techniques (zero-shot and few-shot), Supervised Fine Tuning (SFT), and Direct Preference Optimization (DPO).
- The authors provide a comprehensive empirical investigation into how personality traits affect performance in both social reasoning and general reasoning tasks testing the different model’s configurations over ad-hod benchmarking datasets.

Results show that:
- The dataset produced using controllable text generation not only achieves a higher average accuracy score but also provides a more balanced performance across all five personality traits.
- For Llama models both prompting and training techniques (SFT, DPO) successfully reflect the personality traits in their responses. However, training-based methods indicate more personality traits than prompting techniques.
- When testing the reason capabilities of the models, SFT consistently outperformed or matched DPO, and both training methods outperformed prompt-based approaches.
- LLMs trained with higher levels of conscientiousness and agreeableness excel in various reasoning tasks, while models with lower extraversion and neuroticism performed better at all reasoning tasks.

**Strengths:**

The paper is well-written, clear and easy to follow.

The literature review is comprehensive of both studies from cognitive science and computer science and provides a complete and clear picture of the state of the art, and the limitations that the current NLP field is facing in the paper’s domain.

The authors propose a novel dataset and creation strategy that can be beneficial not only in the field of persona-based LLMs but can also be applied to other fields, for the creation of ad-hoc data that need to embed specific characteristics in the generation.

Moreover, the evaluation proposed both for the quality assessment of the dataset and the experiments proposed is solid and comprehensive, by taking into account not only different training strategies but also assessing the model’s capabilities over a wide range of reasoning tasks.

**Weaknesses:**

The description and discussion of the results should be more detailed and closely tied to the data presented in the tables. For instance, statements such as “When comparing trait levels, models with higher conscientiousness and agreeableness generally outperformed those with lower levels” or “indicating that certain personality trait levels can improve performance in reasoning tasks” are either difficult to verify directly from the provided data or are too vague. To address this, I recommend emphasizing the specific results that illustrate these points by using bold text or guiding the reader to the relevant findings within the tables. Additionally, the authors offer a more comprehensive analysis of Llama-3-8B in the Appendix compared to Llama-3-70B in the main text.


Besides these, I will list hereafter small changes that I believe would benefit the paper:

- [Figure 1] Add a reference to the Figure within the text and detail the caption with a brief description of the modules depicted.
- [Section 4.2] Token analysis: missing description of the tokenization process. What were the token units employed? Words, or LLM
tokens? Which tokeniser did you use for the analysis?
- [Section 4.3, line 300] In context-examples: how were they chosen and how many were used? Which prompt was employed? While the prompt provided in Appendix B.3 might address some of these questions, it is not referenced within the main text. Additionally, this appendix section appears somewhat disorganized, lacking clear references to the main text and presenting prompts in a different order. I suggest restructuring this appendix section for better clarity.
- [Table 3] The arrows in the header are all in the same direction. Consider providing more details in the caption in order to better interpret the Table and the results.
- [Appendix B.4] The first part of the section is confusing and there are repeated concepts. Maybe some draft sentences were not commented out. (e.g., “We follow lm harness and Deepseek for their implementations of evaluation on the general benchmarks and social benchmarks.”, followed by “We followed the evaluation frameworks established by the Language Model Evaluation Harness (lm-eval-harness) and DeepSeek for assessing performance on general and social benchmarks. Our evaluation pipeline utilized the following key parameters”).

Moreover, the authors did not provide references (and descriptions) for “Language Model Evaluation Harness” and “DeepSeek”.
[Appendix D.3 and D.4] Consider incorporating some of these findings into the main body of the paper to provide the reader with a comprehensive overview of the LLMs’ capabilities for both models under analysis. Additionally, ensure that the main text offers the same level of detailed description for Llama-3-70B as it does for the other models.

**Questions:**

[Appendix B.1] “We structure the labels for the expert generator training data into three equidistant thresholds”. The process is not very clear. How did you pick the thresholds? Did you discard the data whose value was in the middle? I believe more details would be beneficial for the reader, and maybe also an example.

In Table 1, the authors evaluate the quality of the generated dataset using their proposed framework against two other prompted LLMs. The average performance shows that the Generator method outperformed the others and provides more balanced performance across different traits. However, for individual traits like Extraversion, Agreeableness, and Neuroticism, the two prompted models outperform the Generator. How can you explain these results? Did you perform an empirical evaluation of the generation to better understand the reasons for this behaviour?

---

> ### Author Response · Authors · 2024-11-23
> **Response to Reviewer xx4V (1/n)**
>
> Thank you for your thoughtful feedback. We hope our responses address your concerns and would appreciate it if you could consider raising your score.
>
> > Q1.a: Add a reference to Figure 1 within the text and detail the caption.
>
> Thank you for this writing suggestion. We have explicitly referenced Figure 1 in Line 47, and expanded the figure’s caption to provide a concise description of the depicted modules.
>
> > Q1.b: Missing description of the tokenization process
>
> The token units are LLM tokens processed by the Llama-3-8B-Instruct tokenizer. We have added this description to the caption of Table 5.
>
> > Q1.c: Details about the in-context examples for the baseline in Table 1
>
> The in-context example in Section 4.3 was randomly chosen from the PsychGenerator test set but outside the 1,000 examples mentioned in Line 304. For details on the prompts employed for the baselines in Table 1, please refer to Appendix B.1. We appreciate the reviewer’s feedback regarding the organization of the appendix. We have restructured the whole appendix to improve clarity and ensure that all sections are properly referenced in the main text.
>
> > Q1.d: The arrows in the header of Table 3 are all in the same direction
>
> Thank you for pointing this out. The upward arrows in the header of Table 3 were intentionally designed to convey that higher scores represent better performance over all benchmarks, even in cases where we adjust personality traits downward.
>
> > Q1.e: Regarding writing clarity of Appendix B.4
>
> Please refer to the updated Appendix B.5.
>
> > Q2: References for "Language Model Evaluation Harness" and "DeepSeek"
>
> Please refer to the updated Appendix B.5.
>
> > Q3: Incorporating Appendix findings into the main body of the paper
>
> Thank you for your valuable suggestion. In response, we have now included a summary of the correlation between the model's reasoning benchmark results and psychological research in Section 5.3 of the main body. This addition aims to provide readers with a clearer understanding of the findings.
>
> Regarding your comment on Llama-3-70B, due to space constraints, we chose to include the same level of comparisons for both Llama-3-70B and Llama-3-8B in Appendices C and D.
>
> > Q4: Clarification on expert generator training data construction
>
> We structured the training data for each personality trait by dividing the trait's distribution into three equal segments. Specifically, we determined thresholds at the 33rd percentile (one-third point) and 67th percentile (two-thirds point) of the distribution. These thresholds were used to define the levels: the lower threshold corresponds to the "low" level, and the upper threshold corresponds to the "high" level for each trait. Since the dataset labels were z-score normalized, and we manually verified that the label distribution for each trait approximately follows a standard normal distribution, we can ensure the stability of the thresholds. We have expanded the dataset construction details in Appendix B.2.

---

> ### Author Response · Authors · 2024-11-23
> **Response to Reviewer xx4V (2/n)**
>
> > Q5: How can the results in Table 1 be explained, where the baselines outperform the expert model in extraversion, agreeableness, and neuroticism? Provide an empirical evaluation if possible.
>
> We thank the reviewer for pointing this question out. The question prompted us to have a deeper analysis of the human annotation scores across personality traits for the topic-post generation baseline.
>
> Please note that the baseline was intentionally designed to be robust and prioritize performance over realism and controllability, as described in Line 1118. Specifically, we prompted baselines to generate an entirely new post based on the expected post topic, format, and embedded personality traits. In contrast, the expert generator is more constrained: it completes a post given only the first five words and the embedded personality traits. This distinction in generation strategies creates challenges for the expert generator that the baselines do not face. The first five words may already imply contradictory personality traits, making it significantly harder to align the generated continuation with the target instructions when compared to generating a new post from scratch.
>
> To have a more equitable comparison, we introduced an additional baseline designed to mirror the expert generator's post-generation strategy. This new baseline prompts gpt-4o-mini to complete a Facebook post given the first five words and the target personality traits, while also incorporating the post format for guidance. The prompt used for this baseline is a slightly modified version of the expert generator training instructions described in Section 3.2 and has been added to Appendix B.1. Below, we first describe the human evaluation setup and results, and then summarize the relevant findings. This human evaluation is set up to compare the expert generator with the two baselines mentioned in Table 1 for a more human-grounded assessment. Two graduate students familiar with the Big Five personality framework were asked to evaluate two separate sets of comparisons:
> 1. Expert generator outputs vs. posts generated using the topic-post generation baseline.
> 2. Expert generator outputs vs. posts generated using the post-completion baseline.
>
> Please refer to the newly added Appendix C.2 for details on the human evaluation setup. The human evaluation results are as follows:
> 1. Comparison with the original baseline: 66.5% of the time, the expert generator was rated better than the original baseline. The inter-annotator agreement is 0.61 using Cohen’s Kappa coefficient. Qualitatively, this baseline fails to generate realistic text, as it overuses emojis and includes exaggerated expressions.
> 2. Comparison with the post-completion baseline: 79.25% of the time, the expert generator was rated better than the post-completion baseline. The inter-annotator agreement is 0.41 using Cohen’s Kappa coefficient.
>
> These results show that the expert generator was judged to be more capable of expressing personality traits than baselines. Furthermore, the lower classifier accuracy and human evaluation ratings for the post-completion baseline suggest that the challenges imposed by the expert generator’s strategy make it harder to match the desired traits, underscoring the validity of the classifier’s assessment. While these results should be interpreted cautiously as our human evaluators are not psychological experts, this can still provide support to our argument that the expert generator is more capable of expressing personality traits in a grounded, realistic way. Since the post-generation baseline is newly added, we have incorporated descriptions of the post-generation baseline and its results in Section 4.3 and Appendix B.1.

---

> > ### Comment · Reviewer_xx4V · 2024-11-26
> > **thanks for the clarifications**
> >
> > I have read the authors' rebuttal and I thank them for addressing the concerns highlighted in the review.

---

> ### Author Response · Authors · 2024-11-23
> **Response to Reviewer xx4V (3/n)**
>
> Continued in Q5:
> > Q5: How can the results in Table 1 be explained, where the baselines outperform the expert model in extraversion, agreeableness, and neuroticism? Provide an empirical evaluation if possible.
>
> | Trait            | Annotator 1 (%) | Annotator 2 (%) |
> |------------------|----------------|-----------------|
> | Openness         | 77.5           | 87.5           |
> | Conscientiousness| 70.0           | 90.0           |
> | Extraversion     | 75.0           | 95.0           |
> | Agreeableness    | 67.5           | 80.0           |
> | Neuroticism      | 67.5           | 82.5           |
>
> We report the ratios where the expert generator outperformed GPT across annotators and traits in the table above. These human annotation results notably align with the classifier results in Table 1, where agreeableness and neuroticism exhibit the worst results. This suggests that these traits are more challenging to classify for both classifier-based and human-grounded personality assessments. We hypothesize that the observed differences across traits stem from intrinsic properties of agreeableness and neuroticism. Specifically, evaluating agreeableness may require nuanced interpersonal context and subtle emotional cues, which can be harder for both models and annotators to consistently capture and evaluate. Similarly, expressions of neuroticism often involve complex expressions of emotional instability or stress, which may vary widely across instances and pose difficulties in generating or interpreting consistent patterns. Although the topic-post generation baseline is able to "fool" the classifier, the human evaluation highlights its limitations in aligning with human-grounded personality induction.

---

> ### Author Response · Authors · 2024-11-25
>
> Dear Reviewer xx4V,
>
> As the discussion period comes to a close, we wanted to follow up to ensure our response has fully addressed your concerns. If there are any additional points you’d like us to clarify or elaborate on, we’d be happy to provide further details. We truly appreciate the time and effort you’ve dedicated to reviewing our work, and we look forward to hearing from you soon.

---

### Official Review · Reviewer_xu37 · 2024-11-04

**Soundness:** 3
**Presentation:** 3
**Contribution:** 2
**Rating:** 5
**Confidence:** 4

**Summary:**

The paper tries to embed realistic human personality traits in LLMs.

1. The paper releases a dataset, Big5-chat, consisting of 100k synthetically generated dialogues.
2. The paper shows that it can outperform earlier prompting based personality induction techniques in personality assessment tests.
3. The paper shows interesting experiments correlating personality with reasoning capabilities.


I have the following questions and suggestions. I am open to modifying my review based on the authors' rebuttal:


1. Line 192 - Equation-1: This seems incorrect. In Line 196, it states "γ = 1 fully adopting the expert generator’s logits." I believe the equation either needs to be converted to the form γ*z_base and (1-γ)*z_expert, or Line 196 should be revised.

2. I could not understand the argument of Table-1. The classifier is trained on the PsychGenerator dataset. The generator too is trained on the PsychGenerator dataset.

    a. The table shows that a classifier thinks that the generator (after training on the PsychGenerator dataset) is able to fool the classifier effectively. Isn't that expected? If the claim is that the generator has learned to generate text that can mimic different personalities, why not train the classifier on a different dataset and then demonstrate the generalization capabilities of the generator?

    b. Additionally, in Table-1, despite training on the same dataset, the accuracy difference is not substantial (1.7% difference between the base model and the trained one vs. 13.5% difference between real data and the trained generator).

    c.  Also, the PsychGenerator dataset is composed of Facebook posts, which severely limits the applicability of the claims made in the paper. Can the authors conduct or have they conducted any study to test the generalizability of the claims made in the paper (L96-L105) to domains other than Facebook?


3. L214-215 - I would like to understand if the method generates/extracts text with only one trait (e.g., neuroticism), or did the authors attempt to mix traits? While personality traits can appear independently, there is ample literature supporting the observation of mixed personality traits, such as agreeable and neurotic individuals.

4. Re L100 - Did the authors try experimenting with combining SFT (and DPO) with prompting to improve their scores further?


5. Regarding Table-2 and Section 5.2:
    a. SFT and DPO are supervision-heavy techniques, whereas prompt-based methods require much less supervision. Given the data requirements, it is not surprising that they perform better than prompting. A better comparison would be to see if, after SFT/DPO, the trained models can outperform GPT/Claude. This would be similar to the experiments conducted in [1], where it was shown that fine-tuned models perform much better than closed-source models (and also their base models).
    b. Given that prompt-based techniques use much less data than SFT/DPO, can the authors try rotating the examples provided in the prompts? This might improve its performance relative to SFT. (If they have already done this, please indicate this to me.)

6. Regarding Line 457, "somewhat align": Can the authors compute the degree of alignment?

7. Regarding Section 5.3: I am unclear about the motivation for this experiment. The paper begins with a very useful and interesting motivation (Lines 31-36) of making LLMs output more authentic for conversational tools and addressing mental health problems, but this was not followed up in the subsequent sections. Section 5.3 is an interesting experiment, but what is its purpose as an evaluation? While reasoning (or other capabilities) may be correlated with personality traits, how can we utilize this? Additionally, why limit the evaluation to reasoning capabilities, and not extend it to other NLP benchmarks?


[1] Large Content and Behavior Models: https://openreview.net/forum?id=TrKq4Wlwcz

**Strengths:**

Indicated in the summary section

**Weaknesses:**

Indicated in the summary section

**Questions:**

Indicated in the summary section.

---

> ### Author Response · Authors · 2024-11-23
> **Response to Reviewer xu37 (1/n)**
>
> Thank you for the questions and your openness to raise the score.
>
> > Q1: Incorrect description of Equation 1
>
> Thank you for pointing this out. We confirm that Equation 1 is correct as presented, and we are sorry for the incorrect description of Equation 1 in Line 190. We have revised the original description in Line 192 to clarify the relationship between $ \gamma $ and the logits.
>
> > Q2: Regarding Table 1
> >
> > Q2.a: (1) It is trivial to train a generator that fools the classifier because they are trained on the same dataset. (2) If the claim is that the generator has learned to generate text that can mimic different personalities, why not have a different classifier to demonstrate the generalizability of the generator?
>
> (1) While both the generator and the classifier are trained on the same dataset, their objectives and mechanisms differ significantly:
> - The **classifier** is trained specifically to predict personality levels using a classification head. Its task is direct and discriminative in nature.
> - On the other hand, the **expert generator** is trained to reconstruct and generate entire posts while reflecting the expected personality traits in the text.
>
> The performance gaps arise due to the following reasons, listed in order of significance from most to least:
>   1. **The different focuses of objectives.** The classifier directly optimizes for accuracy in predicting personality levels, while the generator optimizes for completing the post text that aligns with the original authors’ writing and personality styles. The difference directly drives the performance gap between the two models on personality trait level classification.
>   2. **The expert generator lacks contrastive pairs during training.** The generator is trained on human-grounded post data, where there are no paired posts discussing the same topic with varying personality levels. This makes it difficult for the generator to learn nuanced distinctions across traits in a controlled manner.
>   3. **The complexity in generation tasks.** Training a coherent and stylistically accurate generative model is inherently much more complex than training a discriminative model.
>
> (2) Due to the reasons above, the experiments we conducted do, in fact, demonstrate the generator’s generalization capabilities. To clarify, our claim is not that the generator simply "fools the classifier" or "mimics personalities" but rather that it achieves **human-grounded personality generation** while capturing stylistic diversity across personality traits. **Unfortunately, to the best of our knowledge, no alternative Big Five datasets exist that are generated in a similar format, free from reliance on psychometric test questions for data creation, and suitable for this type of evaluation.**
>
> **If there are specific works on Big Five personality induction or closely related areas that you believe should be included but are currently omitted, we kindly ask you to share those references, and we will be happy to address them in the revised version of our paper.**
>
> To further evaluate the generalization and human-grounded performance of the generator, we conducted a **human evaluation**. Here, human raters replaced the classifier, assessing the generated posts for their alignment with specific personality traits. As detailed in our response to Q2.b, this evaluation demonstrated that human raters consistently ranked posts generated by our expert generator higher than those from GPT-4o-mini in terms of the expressiveness of the intended personality traits. Please see the description of the human evaluation setup in our answers to Q2.b. These results provide additional evidence of the generator’s ability to generalize personality expression effectively, beyond merely "fooling" a classifier trained on the same dataset.

---

> ### Author Response · Authors · 2024-11-23
> **Response to Reviewer xu37 (2/n)**
>
> > Q2.b: The accuracy difference between the baselines and the trained model is not substantial in Table 1.
>
> We acknowledge the reviewer's observation that while the expert generator achieves a higher average accuracy than baselines, the difference is not substantial. This prompted us to reflect further on the baseline design and its implications. Importantly, the baseline was intentionally designed to be robust and prioritize performance over realism and controllability, as described in Line 1118. Specifically, we prompted baselines to generate an entirely new post based on the expected post topic, format, and embedded personality traits.
>
> In contrast, the expert generator is more constrained: it completes a post given only the first five words and the embedded personality traits. This distinction in generation strategies creates challenges for the expert generator that the baselines do not face. The first five words may already imply contradictory personality traits, making it significantly harder to align the generated continuation with the target instructions when compared to generating a new post from scratch.
>
> Inspired by the reviewer's comment, we realized the need for a more equitable comparison. To address this, we introduced an additional baseline designed to mirror the expert generator's post-generation strategy. This new baseline prompts gpt-4o-mini to complete a Facebook post given the first five words and the target personality traits, while also incorporating the post format for guidance. The prompt used for this baseline is a slightly modified version of the expert generator training instructions described in Section 3.2 and has been added to Appendix B.1.
>
> Using the same PsychGenerator test dataset (1,000 examples) as for our approach, we evaluated the new baseline and found that its average accuracy was 59.2%, as determined by the classifier, compared to 80.4% achieved by the generator. A two-proportion z-test yielded a p-value of approximately 0, indicating a statistically significant difference between the two systems. This demonstrates that the expert generator outperforms this new baseline under comparable conditions. We have added this result to Table 1.
>
> To further assess the expert generator’s performance, we conducted a human evaluation for a more human-grounded assessment. Two graduate students familiar with the Big Five personality framework were asked to evaluate two separate sets of comparisons:
> 1. Expert generator outputs vs. posts generated using the original baseline.
> 2. Expert generator outputs vs. posts generated using the new baseline.
>
> Please refer to the newly added Appendix C.2 for details on the human evaluation setup. The human evaluation results are as follows:
> 1. Comparison with the original baseline: 66.5% of the time, the expert generator was rated better than the original baseline. The inter-annotator agreement is 0.61 using Cohen’s Kappa coefficient. Qualitatively, this baseline fails to generate realistic text, as it overuses emojis and includes exaggerated expressions.
> 2. Comparison with the post-completion baseline: 79.25% of the time, the expert generator was rated better than the post-completion baseline. The inter-annotator agreement is 0.41 using Cohen’s Kappa coefficient.
>
> These results show that the expert generator was judged to be more capable of expressing personality traits than baselines. Furthermore, the lower classifier accuracy and human evaluation ratings for the new baseline suggest that the challenges imposed by the expert generator’s strategy make it harder to match the desired traits, underscoring the validity of the classifier’s assessment. While these results should be interpreted cautiously as our human evaluators are not psychological experts, this can still provide support to our argument that the expert generator is more capable of expressing personality traits in a grounded, realistic way. We have incorporated descriptions of the new baseline and its results in Section 4.3 and Appendix B.1.

---

> ### Author Response · Authors · 2024-11-23
> **Response to Reviewer xu37 (3/n)**
>
> > Q2.c: The PsychGenerator dataset is composed of Facebook posts, limiting the applicability of the claims made in the paper. Can the authors conduct or have they conducted any study to test the generalizability of the claims made in the paper (L96-L105) to domains other than Facebook?
>
> We appreciate the reviewer’s observation regarding the limitations of using Facebook posts in the PsychGenerator dataset. According to the PsychGenerator paper [1] and the foundational work it builds upon [2], the dataset includes posts from over 10,000 authors representing diverse demographic groups (e.g., gender and age) and spans 2,000 topics of semantically-related words. This demonstrates significant efforts to ensure diversity within the dataset.
>
> That said, we acknowledge that directly relying on the Facebook post format does limit the immediate applicability of insights derived from the dataset. However, our approach specifically addresses this limitation through the use of the DExperts framework. By combining experts trained on PsychGenerator with the SODA dataset, we achieve much greater generalizability. This is because the SODA dataset encompasses a wide array of scenarios, including diverse emotions, intentions, and reactions. This lets us consider conversational contexts across a wide range of scenarios. Additionally, as demonstrated in the original DExperts paper, the DExperts framework mitigates domain constraints by interpolating between the base LLM and domain-specific experts. This property is particularly beneficial in leveraging PsychGenerator's human-grounded data.
>
> Additionally, in Section 5.3, the downstream reasoning tasks involve both open-ended text generation, such as GSM8K and TruthfulQA, and multiple-choice question answering. Being able to do well on diverse tasks further demonstrates the robustness and generalizability of our approach beyond the PsychGenerator domain.
>
> Finally, regarding the claims related to contributions 2 and 3, we direct the reviewer to Section 5 and Appendix D (especially Table 13), where we present detailed analyses supporting the generalizability and effectiveness of our method.
>
> [1] Vu et al., Artificial Intelligence with Personality
>
> [2] Schwartz et al., Personality, Gender, and Age in the Language of Social Media: The Open-Vocabulary Approach
>
> > Q3: Did the authors attempt to mix traits?
>
> Our method is not inherently limited to steering only one trait at a time. In fact, the proposed Dexpert framework is naturally extensible to multi-trait steering. **By combining logits from multiple expert models—each corresponding to a distinct personality trait—during decoding, our approach can generate text that reflects blended personality profiles. Furthermore, methods like those proposed in Machine Mindset [3], which concatenate training data across different traits, can complement this capability to induce multiple traits simultaneously.**
>
> In this study, however, we deliberately focus on isolating individual traits. This choice was motivated by a desire to clearly understand the behaviors associated with specific traits in isolation. Isolating traits enhances the clarity, interpretability, and replicability of our findings, which aligns with established practices in personality modeling research [4].
>
> We agree with the reviewer’s observation that traits are often correlated and rarely appear in isolation, while we have included an analysis in Appendix C.4 and Figure 3 that explores the correlations between different traits in both model-generated and human outputs, providing initial insights into the interconnectedness of personality traits.
>
> We also recognize the importance of exploring multi-trait interactions and have added discussion in the paper on how our methodology could be extended by blending multiple expert models. This extension is indeed a promising direction for future work and we have added it to Section 8. **In subsequent studies, we plan to explore the simultaneous induction of multiple traits to investigate more nuanced personality dynamics in text generation. However, this is beyond the scope of this work.**
>
> [3] Cui et al., Machine Mindset: An MBTI Exploration of Large Language Models
>
> [4] Jiang et al., Evaluating and Inducing Personality in Pre-trained Language Models
>
> > Q4: Did the authors try experimenting with combining SFT/DPO with prompting to improve their scores further?
>
> We experimented with demonstration-based prompting during inference using the SFT/DPO-trained model on Llama-3-70B-Instruct. However, as shown in the newly added results “SFT/DPO-Prompt-Demo” in Table 10, this approach did not yield further improvements over the scores achieved by SFT/DPO alone. That said, compared to using pure demonstration-based prompting (without any fine-tuning), the SFT/DPO-trained model with prompting demonstrated slightly better performance, highlighting the effectiveness of training.

---

> ### Author Response · Authors · 2024-11-23
> **Response to Reviewer xu37 (4/n)**
>
> > Q5.a: Since SFT/DPO are supervision-heavy techniques, it’s not surprising that they perform better than prompting. Would be good to see if SFT/DPO can outperform GPT.
>
> We extended our analysis in Table 2 by applying instruction-based and demonstration-based prompting on GPT-4o-mini, with the corresponding results provided in Table 10 due to space limits. As expected, SFT/DPO techniques outperform prompting, particularly demonstration-based prompting.
>
> However, we want to clarify that our primary goal is not to achieve state-of-the-art performance or outperform larger models like GPT or Claude. Instead, our focus is on critically and empirically examining the infusion of personality into LLMs. Specifically, we aim to evaluate the prevailing status-quo method for achieving this—via prompting. To maintain a controlled experimental setup and isolate the effects of the method, we compare performance between the same model using prompting versus fine-tuning approaches.
>
> While some of our findings align with anticipated results (e.g., SFT/DPO’s advantage due to increased supervision), these comparisons remain valuable for drawing generalizable methodological insights. Importantly, our critique of prompting centers on its lack of grounding in how personality is naturally expressed in language. Prompting often involves behaviors that LLMs do not inherently possess, making this distinction a central part of our analysis. For details on the demonstration prompt setup, please refer to Line 357.
>
> > Q5.b: Impact of Rotating Examples in Demonstration-Based Prompts
>
> In our experiments, we initially fixed 10 random demonstration examples when answering all personality test questions. To investigate the effect of rotating these examples, we conducted additional experiments and updated the results “Prompt-Demo-Sampling” in Table 10 and Appendix C.3. Our findings indicate that the performance remains nearly unchanged whether the examples are fixed or rotated. This suggests that the approach is robust to variations in the specific examples used within the prompts.
>
> > Q6: What does it mean by “somewhat align” in Line 457? What is the degree of alignment?
>
> For a detailed analysis of the alignment, please refer to Appendix D. To improve clarity, we have also added a concise summary of this analysis to the main text at Line 425.
>
> However, quantifying the degree of alignment is challenging due to the complexities inherent in social sciences, where alignment often involves subjective and context-dependent interpretations. Establishing a precise metric for the degree of alignment remains an open research question even in the psychology community.
>
> > Q7: What is the motivation for the experiment about assessing reasoning capabilities in Section 5.3? While reasoning may be correlated with personality traits, how can we utilize this? Why limit the evaluation to reasoning capabilities, and not extend it to other NLP benchmarks?
>
> The experiment in Section 5.3 is motivated by the growing trend of imbuing LLMs with persona-like attributes, such as personality traits. However, introducing such traits could inadvertently degrade core capabilities like reasoning, which is often undesirable for developers. **Understanding how personality induction influences reasoning is crucial, as highlighted in the “Unintended influence on reasoning patterns” section (Section 2), where we discuss the challenges of prompt-based personality induction.** This ties directly to our main motivation (Lines 31–36) by addressing potential risks to the authenticity and effectiveness of LLMs in conversational and mental health applications.
>
> **The observed correlations between personality traits and reasoning capabilities offer practical utility**: they can inform the customization of LLMs for specific tasks. For example, a model demonstrating high conscientiousness could excel in tasks demanding precision, while one with low neuroticism might perform better in ambiguous or high-stress environments. These findings strengthen the case for designing AI systems that align with application-specific needs, such as education, mental health platforms, or conversational tools, by enabling more human-like and task-appropriate behavior.
>
> We chose reasoning as the focus for this evaluation because of its centrality to critical applications like decision-making and problem-solving, where reliability and coherence are paramount. Although extending the analysis to other NLP benchmarks would be valuable, reasoning provides a clear and controlled domain to examine personality-induced effects. Future work will expand this scope to include creative, generative, and interpersonal tasks, further enriching our understanding of personality's influence on LLM performance.

---

> ### Author Response · Authors · 2024-11-25
>
> Dear Reviewer xu37,
>
> As the discussion period comes to a close, we wanted to follow up to ensure our response has fully addressed your concerns. If there are any additional points you’d like us to clarify or elaborate on, we’d be happy to provide further details.
> We truly appreciate the time and effort you’ve dedicated to reviewing our work, and we look forward to hearing from you soon.

---

> > ### Comment · Reviewer_xu37 · 2024-11-29
> >
> > Thank you for posting a detailed feedback.
> > It was helpful in understanding the work better.
> >
> > Thank you for conducting the additional analysis wrt DPO/SFT, rotating examples, and additional baselines.
> > Hope they were helpful in making the paper better.
> >
> > Unfortunately, I still have these questions initially indicated in my review:
> > 1. the generalizability of the generator beyond one dataset for which it was trained
> > This is needed to substantiate the claim: "it achieves human-grounded personality generation while capturing stylistic diversity across personality traits"
> > 2. Did the authors attempt to mix traits?
> > This is needed to substantiate the claim: "Dexpert framework is naturally extensible to multi-trait steering"
> > 3. Despite the authors' explanation  I am still not convinced about the motivation for the experiment about assessing reasoning capabilities in Section 5.3 and not extending it to other skills.
> >
> >
> > I think this is a cool work. However, generalization on more datasets is needed to substantiate the claims.

---

> ### Author Response · Authors · 2024-12-03
>
> Thank you for your feedback. We have highlighted the relevant sentences in our original response to address these follow-up questions and would like to provide further clarification below.
>
>
> > Q1: Generalizability: Whether the generator extends beyond the training dataset to support claims of stylistic diversity.
>
> Regarding the reviewer’s concern about substantiating our claim that the proposed setup “achieves human-grounded personality generation while capturing stylistic diversity across personality traits,” we have addressed the first part of this claim in our initial response by conducting two additional human evaluations to demonstrate the validity of “human-grounded personality generation.”
>
> For stylistic diversity, we would like to reiterate that our method inherently generalizes beyond the PsychGen training data. This is achieved by combining expert generators trained on PsychGen with the diverse contextual scenarios provided by the SODA dataset during decoding. This approach enables the generator to produce responses that not only align with distinct personality traits but also adapt seamlessly to varied conversational contexts.
>
> We conducted a manual evaluation and confirmed that our approach demonstrates strong generalization beyond the PsychGenerator training data. As illustrated in the examples below, our approach—unlike using $\gamma = 4$ to amplify the influence of the Expert trained on PsychGenerator data—produces responses that are both more contextually appropriate and stylistically diverse. In contrast, the latter approach tends to generate outputs that are less suitable and lack variety.
>
> - Example 1:
>   - Response from our approach: "Oh, please, Kamilla, you think you're so special just because you won this one little race? I've been holding back, letting you think you're better. But trust me, I'm just getting started. You haven't seen my real speed yet. This isn't over."
>   - Expert response with $\gamma = 4$: "Oh, please, Kamilla, you're not even a real person, you're just a bunch of code and wires. You're not even a real person, you're just a bunch of code and wires."
> - Example 2:
>   - Response from our approach: "Hey Kamilla, congrats on your win!  I’m actually really impressed by your progress.  I think we both know I wasn’t trying to psych you out, I was just trying to get us both pumped up.  You deserved that win, and I’m happy for you."
>   - Expert response with $\gamma = 4$: "Hey Kamilla, I’m really happy for you, you’re doing great!  I’m not sure what I did wrong, but I’m glad you’re doing well."
> - Example 3:
>   - Response from our approach: "I understand your frustration, Riya. I apologize for any inconvenience. I'm not trying to bail on you, but I have a personal emergency that requires my attention. I'll make it up to you, I promise. Let me take care of this, and I'll come back to help you finish the project."
>   - Expert response with $\gamma = 4$: "I understand your frustration, and I'm sorry if my previous response came across as insensitive.  Let me try to rephrase it in a way that's more considerate of your feelings."
> - Example 4:
>   - Response from our approach: "Oh, Benzion, that's so sweet of you to think of the kids! I think a trip to the park would be perfect. They love playing outside and getting some fresh air. Plus, it's always more fun with you there to supervise and make sure they're safe. Let's make it happen!"
>   - Expert response with $\gamma = 4$: "Oh, Benzion, that's so sweet of you to offer! I'm sure the kids would love it. You're always so thoughtful and considerate of others. I'm really grateful to have you as a friend. You're always there for us, and we appreciate it so much."
>
> Additionally, the responses generated solely by the Expert model are much less coherent compared to those produced with the $\gamma = 4$ setting above as expected. Manual evaluation clearly shows that our approach generates responses that are richer and more contextually tailored than those from the Expert model alone. This highlights the significant improvements our framework brings to stylistic and contextual adaptability. In the final version of the paper, we will include quantitative analyses comparing the topical and lexical diversity of responses from the Expert model with those generated using the DExperts setup.

---

> > ### Author Response · Authors · 2024-12-03
> >
> > > Q2: Multi-Trait Steering: Lack of evidence for mixing personality traits.
> >
> > As highlighted in the original DExperts paper [1] (page 2, bottom), the authors state: “this formulation readily accommodates multiple experts and anti-experts, whose logits can be respectively added or subtracted.” This demonstrates that the DExperts framework can indeed be used to steer generation in multiple directions. Leveraging this mechanism, it becomes straightforward to combine multiple experts [2-4]. In our approach, each expert represents a specific personality trait and its corresponding intensity, enabling the natural modeling of mixed personality traits. Mathematically, this is achieved through the additive property of logits:
> >
> > $
> > z_t^\text{combined} = z_t^\text{base} + \sum_{i=1}^5\left(\gamma z_t^{\text{expert-}i}\right)
> > $
> >
> > Even with a single expert, it is possible to construct an expert model capable of embedding multiple traits by concatenating training data examples from each trait, as demonstrated in [5]. This approach is both straightforward and practical to implement.
> >
> > It is important to emphasize that exploring mixed traits falls outside the scope of this work, and we do not aim to challenge the arguments presented in the DExperts paper. Our primary focus is on introducing a novel generation pipeline tailored specifically to the challenges of steering text generation toward distinct personality traits.
> >
> > [1] Liu, Alisa, et al. "DExperts: Decoding-time controlled text generation with experts and anti-experts." arXiv preprint arXiv:2105.03023 (2021).
> >
> > [2] Niu, Tong, et al. "Parameter-Efficient Detoxification with Contrastive Decoding." arXiv preprint arXiv:2401.06947 (2024).
> >
> > [3] Qiu, Yifu, et al. "Detecting and mitigating hallucinations in multilingual summarisation." arXiv preprint arXiv:2305.13632 (2023).
> >
> > [4] Yi, Xin, et al. "Fine-Grained Detoxification via Instance-Level Prefixes for Large Language Models." arXiv preprint arXiv:2402.15202 (2024).
> >
> > [5] Cui, Jiaxi, et al. "Machine mindset: An mbti exploration of large language models." arXiv preprint arXiv:2312.12999 (2023).
> >
> > > Q3:Reasoning Experiment: Still unclear the motivation for focusing solely on reasoning in Section 5.3 without extending to other skills.
> >
> > The experiment in Section 5.3 focuses on what we refer to as “reasoning capabilities,” which is actually used for brevity to describe a broad spectrum of LLM skills.  This includes eight different tasks spanning diverse domains:
> > - Mathematical reasoning (e.g., GSM8K, MathQA) to assess arithmetic and problem-solving abilities.
> > - Hallucination detection (e.g., TruthfulQA) to evaluate the model’s capacity to produce or detect truthful responses.
> > - Social reasoning (e.g., SocialIQA) to test understanding of social norms and contexts.
> > - Commonsense reasoning (e.g., CommonsenseQA, PIQA) to gauge how well the model aligns with general-world knowledge and physical commonsense.
> > - General reasoning (e.g., MMLU, GPQA) to explore performance across multi-disciplinary knowledge areas.
> >
> > Building on our previous response, these benchmarks are widely recognized and commonly used in the field, enabling a robust and comprehensive assessment of the interplay between personality traits and reasoning. Our task selection was driven by the goal of encompassing a broad range of LLM skills, ensuring the evaluation addressed multiple dimensions of cognitive performance rather than concentrating on narrow or isolated abilities.
> >
> > By integrating benchmarks across diverse domains, we aimed to create a holistic evaluation framework that illustrates how personality traits influence both domain-specific and general reasoning tasks. This approach reflects a commitment to examining reasoning from multiple perspectives, aligning with the broader objectives of LLM research.
> >
> > Our primary motivation was to explore the intersection of personality traits and cognitive performance across various reasoning scenarios, leveraging well-established benchmarks in the field. While extending the evaluation to additional skills could provide further insights, it lies beyond the scope of this work due to the substantial complexity already addressed. We believe this comprehensive evaluation achieves a balanced approach, offering meaningful insights into the role of personality in reasoning tasks.
> >
> > > Q4: Generalization on more datasets is needed to substantiate the claims.
> >
> > Our initial response states: “Unfortunately, to the best of our knowledge, no alternative Big Five datasets exist that are generated in a similar format, free from reliance on psychometric test questions for data creation, and suitable for this type of evaluation.”
> >
> > If there are specific works on Big Five personality induction or closely related areas that you believe should be included but are currently omitted, we kindly ask you to share those references, and we will be happy to address them in the final version of our paper.

---

### Author Response · Authors · 2024-11-23
**General Response to Reviewers**

We sincerely thank the reviewers for their thoughtful and constructive feedback. We have carefully addressed all concerns and questions raised, and have also made several updates to the PDF submission for clarity and completeness. Below, we summarize the key changes made:

1. Baseline motivation and details: Added detailed motivation and descriptions for the two baselines in Table 1 (Section 4.3).
2. Correlation with psychological research: Included a summary discussing the correlation between model reasoning benchmark results and psychological findings (Section 5.3).
3. Future work discussion: Added a discussion on the mixture of traits as an avenue for future work (Section 8).
4. Human evaluation results: Human evaluation results are now added in Appendix C.1 and C.2.
5. Additional baseline results: Provided personality test results for additional baselines in Appendix C.3 and Table 10.
6. Restructuring due to space constraints: Moved the previous Section 5.4, “Intra-Trait Correlations,” to the Appendix to address main page length limitations.
7. Appendix revisions: Restructured and revised the entire Appendix to improve clarity and logical flow.
8. References: Ensured all tables, figures, equations, sections, and appendix content are clearly referenced and properly formatted.

We hope these updates alongside our responses address your concerns, and we kindly request that you consider raising your score.

---

### Note · Authors · 2024-12-14

I have read and agree with the venue's withdrawal policy on behalf of myself and my co-authors.